# Evaluating the effects of soil erosion and productivity decline on soil carbon dynamics using a model-based approach

Samuel Bouchoms[1], Zhengang Wang[2], Veerle Vanacker[1], Kristof Van Oost [1]

[1]TECLIM - Georges Lemaître Centre for Earth and Climate Research, Université Catholique de Louvain, Louvain-la-Neuve BE 1348, Belgium;

[2] School of Geography and Planning, Sun Yat-sen University, Guangzhou, 510275, China

*Correspondence to*: Kristof Van Oost (Kristof.vanoost@uclouvain.be)

**Abstract.** Sustained accelerated soil erosion alters key soil properties such as nutrient availability, water holding capacity, soil depth and texture, which in turn have detrimental effects on crop productivity and therefore reduce C input to soils. In this study, we applied a 1-D soil profile model that links soil organic carbon (SOC) turnover, soil erosion and biomass production. We used observational data to constrain the relationship between soil erosion and crop productivity. Assuming no changes in effort, we evaluated the model performance in terms of SOC stock evolution using published observational data from 10 catchments across Europe and the USA. Model simulations showed that accounting for erosion-induced productivity decline (i) increased SOC losses by 37 % on average relative to a scenario where these effects were excluded, and (ii) improved the prediction of SOC losses when substantial soil truncation takes place. Furthermore, erosion-induced productivity decline further reduced soil-atmosphere C exchange by up to 30 % after 200 years of transient simulations. The results are thus relevant for longer-term assessments and they stress the need for integrated soil-plant models that operate at the landscape scale to better constrain the overall SOC budget.

## 1 Introduction

The soil system represents one of the most important carbon (C) pools by storing around 1417 Pg C in the upper first meter. As a result, its impact on the global C cycle and climate has been widely recognized and studied (Hiederer and Köchy, 2011; Houghton, 2007; Crowther et al., 2016). The terrestrial carbon cycle is mainly driven by soil-atmosphere exchanges; vegetation takes up carbon from the atmosphere and provides input into the soil in forms of root excretions and plant residues while biologic activity and in-situ mineralization release carbon from soils back to the atmosphere (Houghton, 2007).

Through vegetation disturbance and agricultural extension, human activities have had an important impact on the soil system, not only by changing the soil C cycle, but also increasing soil erosion rates up to two orders of magnitudes (Vanacker et al., 2013; Gregorich et al., 1998; Montgomery, 2007). Soil erosion affects vegetation growth and biomass production by changing

soil physical and chemical properties related to soil fertility such as water holding capacity, nutrient status or soil depth Bakker et al., 2004). Effects of soil erosion on crop productivity have intensively been studied during the past decades for a wide range of pedological and climatic conditions (Kosmas et al., 2001; Bakker et al., 2004; Fenton et al., 2005; Gregorich et al., 1998). In this study, we consider crop productivity to be directly proportional to the total amount of plant tissues produced (in kg) per

unit of area. As a result, productivity is directly related to biomass productivity. These experimental studies have indicated that for a given agricultural management practice, crop productivity tends to decrease when soil is subject to erosion (Bakker et al., 2004; den Biggelaar et al., 2003; Larney et al., 2016). On the long term, this reduction is expected to result in an additional loss of SOC due to decreasing C inputs in the soils (Gregorich et al., 1998; Doetterl et al., 2016; Kirkels et al., 2014). Although large uncertainties remain about the strength and the form of the relationship between crop productivity and soil erosion, the

general tendencies and underlying mechanisms have been identified through data meta-analyses (e.g. Bakker et al., 2004; Chappell et al., 2012).

In addition to changes in soil C inputs, human-induced erosion also resulted in lateral redistribution of soil particles across the landscapes and subsequent SOC transfer to the fluvial system (e.g. Van Oost et al., 2005a). Soil redistribution by erosion

affects SOC dynamics through changes in physical protection as a result of the breakdown of aggregate structure during transport from eroding sites to deposition sites, (partial) replacement of eroded C by new photosynthates at eroding sites, and burial and enhanced preservation in depositional areas (Stallard, 1998; Harden et al., 1999; Van Oost et al., 2007b; Lal., 2003; Hoffmann et al., 2009; Wang et al., 2014). Although each of these three processes is individually relatively well understood, the result of their interactions at the landscape scale is still poorly constrained (Kirkels et al., 2014). A dynamic representation

of the interactions between soil erosion, crop growth and SOC turnover is needed in order to better constrain the overall C fluxes in eroding landscapes (Chappell et al., 2015; Harden et al., 1999).

During the last years, several coupled soil erosion-C turnover models have been presented: some of them are point models that operate at the soil profile scale (e.g. Billings et al 2010, Harden et al., 1999; Manies et al., 2001). Other are spatially explicit and focus on the representation of geomorphic processes and SOC turnover in a three-dimensional context (e.g. Dialynas et

al., 2016, Fiener et al., 2015, Van Oost et al., 2005, Wilken et al., 2017). They operate at timescales from single events (e.g. tRIBS-ECO, Dialynas et al., 2016; MCST-C, Wilken et al., 2017) to annual (Van Oost et al., 2005) while others (e.g. Vanwalleghem et al., 2013; Rosenbloom et al., 2006; Yoo et al., 2006) focus on long-term landscape evolution. The point-models have a detailed representation of soil-plant systems and are typically based on ecosystem models assuming first-order decay (e.g. Harden et al., 1999; Liu et al., 2003; Lugato et al., 2016). For example, the CENTURY model simulates the

dynamics of C, nitrogen, phosphorus, and sulphur for different plant-soil systems (Parton, 1996) and can be modified to represent erosion-induced C losses or gains (e.g. Harden et al., 1999; EPIC, Izaurralde et al., 2001, 2007). The key advantages of the approach adopted in these studies are that it (i) allows to represent management practices and (ii) to simulate how plant-derived C inputs evolve over time with ongoing erosion. Most models were developed as short-term decision-making tools for agricultural (or grassland) management. These models not only have allowed us to predict the consequence of specific

management options, they also provided insights into the geomorphic soil plant-response at different spatial scales. However, most models were applied to reproduce the temporal evolution of soil-atmosphere C exchange of a specific study site (Manies et al., 2001; Liu et al., 2003) or were applied at larger spatial scales (e.g. Lugato et al., 2016) but without thorough model validation due to the lack of observational data. To our knowledge, few modelling studies addressed how erosion-induced productivity decline influences C turnover and soil-atmosphere C exchange in detail on the long-term. This study proposes a step in that direction by explicitly linking crop productivity, soil properties and SOC dynamics in a soil profile model to explore the longer-term (i.e. decades to centuries) effect of soil erosion on SOC stocks and fluxes. The model accounts for vertical soil-atmosphere C exchange, lateral SOC displacement and C inputs into the soil at the profile scale. Rather than using a process-based soil-plant model, which faces issues such as parameter estimation and model structure selection (e.g. Beven, 2007), we propose a parsimonious approach where erosion-crop productivity relations are implemented based on observational erosion-productivity relations. Our objectives are (i) to evaluate the performance of a parsimonious coupled model by confronting model simulations to available data and (ii) to investigate the longer-term (i.e. centennial) effect of erosion on crop productivity and SOC dynamics at the profile scale.

## 2. Material and methods

### 2.1 Erosion effect on crop productivity: data meta-analysis

To represent the effect of erosion on crop productivity, we opted for an empirical approach based on the dataset of 24 studies compiled by Bakker et al. (2004). This dataset is one of the most comprehensive meta-analysis available and evaluates crop productivity response to soil erosion for a broad set of environmental conditions, crop growth constraints, soil conditions and experimental methodologies. This approach compares plots with different degrees of erosion but similar characteristics in terms of landscape position, slope and management practise. Crop productivity relative to non-eroding conditions were reported by Bakker et al. (2004) where a relative crop productivity of 1 indicates that there is no erosion-induced change in crop productivity, values smaller than one represent crop productivity losses and values larger than 1 crop productivity gains. In their meta-analysis, Bakker et al. (2004) discussed three functional forms of erosion–crop productivity relationships (Fig. 1): a rapid and non-linear decrease of crop productivity, a continuous and linear decrease, and a slow and non-linear decrease (Bakker et al., 2004). The empirical basis is relatively small, and the individual studies used by Bakker et al. (2004) clearly show that local environmental conditions can strongly affect the form of the relationship between soil loss and crop productivity. As a result, a generally applicable response 'model' does not exist. To tackle this issue, we use a broad range of potential trajectories that cover the scatter observed. We explore the full range of constraints form of soil truncation on crop productivity using the following equation:

$$Ydr = -\alpha \, \mathrm{Tr}^B + 1 \qquad\qquad\qquad (1)$$

where *Ydr* is the relative crop productivity, *Tr* is the cumulative soil truncation since the start of cultivation (m), $\alpha$ is the maximum crop productivity reduction and *B* is the power law exponent linking the relative crop productivity to soil erosion.

Based on the analysis of the Bakker et al. (2004) data, α was set to 0.42 (Fig. 1). Note however that only data from comparative-plots were included in our analysis to parametrize the relationship as Bakker et al. (2004) pointed out that this method is the most appropriate to estimate erosion effects on crop productivity. Even though the number of studies is low, we argue that it represents a range of realistic range of trends of crop productivity declines in response to soil erosion. Soil depth was indirectly considered using a clay content profile which is represented as a fraction of the soil volume. It should be noted that the relationships between relative crop productivity and soil truncation described and discussed hereafter assume no differences in agricultural practices between eroding and non-eroding conditions. Hence, there is no specific adaptation in practices or effort to counteract the decline. Furthermore, when assuming that a linear relation between crop productivity and biomass production is reasonable, the relative crop productivity as presented by Bakker et al (2004) are proportional to biomass productivity. We hereafter refer to crop productivity and assume no change in agricultural practices or efforts during our simulations.

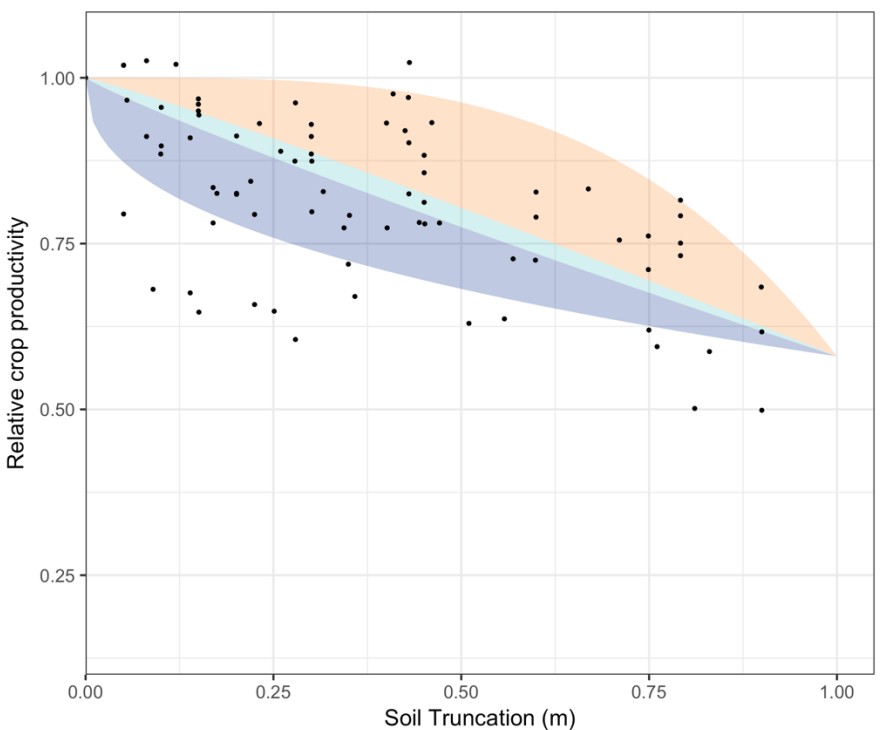

**Figure 1** – Relative crop productivity decrease as a function of soil truncation based on paired-plot experiments. Observations are taken from the data meta-analysis presented by Bakker et al. (2004). Values larger than one indicate a gain in crop productivity and values smaller than one indicate a loss of crop productivity. The three shaded areas represent the space of the relationships investigated in our study. Dark

blue, cyan and orange shades denote respectively the concave relationship (B < 0.9), linear relationship (0.9 < B < 1.1) and convex relationship (B > 1.1).

## 2.2 The SOC turnover model

### 2.2.1 ICBM

Building on existing work, we used a SOC turnover model that is coupled to a dynamic representation of the SOC and clay profiles in response to ongoing erosion (Fig. 2). SOC cycling was represented by a depth explicit version of the Introductory Carbon Fluxes Model (ICBM, Andren and Katterer (1997)) which has been implemented in coupled models (e.g. Van Oost et al., 2005). ICBM is a two-pools carbon model simulating SOC transfer from the roots, residue and manure to a 'young' C

pool, transfer from the 'young' pool to an 'old' C pool and C mineralization in both pools (Andren and Katterer, 1997). The model time step is 1 year. SOC fluxes are described by the following equations:

$$\frac{dY}{dt} = i - k_y \, r \, Y \,, \tag{2}$$

$$\frac{dO}{dt} = h \, k_y \, r \, Y - k_o \, r \, O, \tag{3}$$

Where $Y$ (Mg C ha$^{-1}$) and $O$ (Mg C ha$^{-1}$) are respectively the young and old SOC pools and $k_y$ (yr$^{-1}$) and $k_o$ (yr$^{-1}$) their turnover rates (Andren and Katterer, 1997). $i$ stands for the total carbon input which is the sum of the input from the crops ($ic$) and manure ($im$). The transfer from the young pool to the old pool, calculated at each time step, is proportional to the humification factor ($h$) and the climatic (temperature and moisture) and edaphic conditions which are condensed in the $r$ coefficient (Andren

and Katterer, 1997). Values for $k_y$ and $k_o$ are respectively 0.8 and 0.006 yr$^{-1}$. Note that the quantity *(1-h) $k_y$ r Y* represents the mineralized/respired amount of C leaving the "young" pool.

The humification factor is estimated as follows:

$\quad h(z) = \frac{ic(z)*hc + im(z)*hm}{ic(z) + im(z)} e^{0.0112(cl(z)-36.5)},$ \hfill (4)

Where *ic(z)* and *im(z)* are the C inputs from crop and manure at the depth *z*, *hc* and *hm* the humification coefficient for respectively crops and manure, and *cl(z)* the clay content at depth *z* (%). Humification coefficients equal 0.3 and 0.125 respectively for *hc* and *hm*. At each time step, the humification values are calculated based on the C input from crop and

manure at the considered time step. Only then are the values of the C pools computed.

The climate factor $r$ represents the effect of external factors on SOC decomposition and integrates the effect of temperature and moisture (Andren and Katterer, 1997). For the same local mean temperature, r may vary relatively to local moisture resulting from soil physical characteristics or agricultural practices changes, with higher value of $r$ denoting moister conditions. The $r$ parameter directly affects $h$ and the mineralized C (Eq.2 and Eq. 3). It was calibrated based on Swedish climate conditions and for bare fallows, for which it was set at 1 (Andren and Katterer, 1997). To adjust the $r$ parameter to the local climate, we used a $Q_{10}$ relationship based on temperature following the recommendation of Andren and Katterer (1997):

$$r = 2.07^{\frac{T-5.4}{10}}, \tag{5}$$

Where $T$ is the mean annual temperature (°C). This relationship reflects the effects of both soil temperature and moisture on decomposer activity.

The model is depth-explicit and considers a depth-dependent C input and mineralization rate (Nadeu et al., 2015; Van Oost et al., 2005b; Wang et al., 2014). While manure and residue-derived C input only affect the topsoil layers, the carbon input from plant roots is distributed throughout the soil profile using the following relationship:

$$\varphi(z) = \begin{cases} 1, z \leq z_r \\ \exp(-\delta(z - z_r)), z > z_r \end{cases}, \tag{6}$$

With $\varphi(z)$ the root density profile from which C input from roots are derived at depth $z$, $z$ is given in meters, $z_r$ is the depth of the topsoil where ploughing is assumed to homogenize the SOC content and $\delta$ is the root density coefficient. The model does not consider bioturbation nor chemical leaching as, in agricultural landscapes, these processes have a relatively low impact on SOC fluxes at the considered timescales here, relative to soil redistribution (Doetterl et al., 2016, Kirkels et al., 2014, Minasny et al., 2015). Furthermore, the model assumes no biological adaptation of the plants whereby roots density and distribution could evolve in reaction to soil erosion. Hence, the root-depth density profile does not change over time.

The turnover rates of the SOC pools as a function of depth are computed as an exponential function:

$$k_{tz} = k_{t0} \exp(ur\,z), \tag{7}$$

Where $ur$ is a dimensionless coefficient of depth attenuation, $k_{t0}$ (yr⁻¹) is the turnover rate at the soil surface and $k_{tz}$ (yr⁻¹) represents the SOC turnover rate at depth $z$. The function applies to both $k_y$ and $k_o$.

The model starts with prescribed SOC and clay content profiles. Carbon turnover is then coupled to the clay content profile through a depth-dependent humification factor (Eq.4). Crop productivity is updated each year following Eq. 1, in relation to the cumulative soil truncation. Crop productivity affects the SOC content by multiplying the *ic* term of Eq. 4 (C input from crop: residues and roots) by the relative crop productivity obtained by Eq. 1. C input from manure (*im*, Eq. 4) is not affected by crop productivity change. Under the absence of site-specific data, we here assume a linear relationship between crop productivity and soil C inputs following Eq. 8:

$$i(t) = i(0)\ Ydr(t)\,, \tag{8}$$

Where *i(t)* is the C input from crop at the time *t*, *i(0)* the initial C input and *Ydr* the relative crop productivity at time *t* compared to initial crop productivity. The implications of this assumption are discussed further.

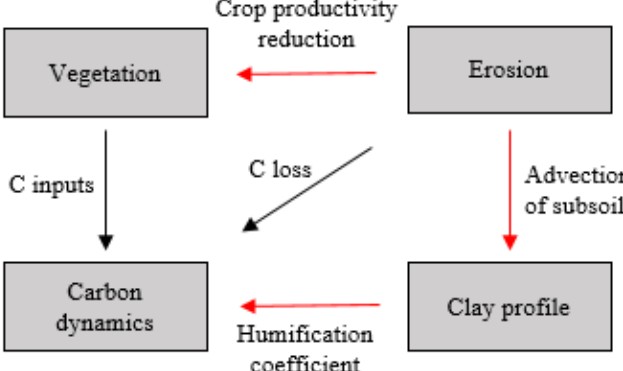

**Figure 2** – Schematic representation of the model. Black arrows depict processes included in published versions of the model (Nadeu et al., 2015) and red arrows represent the new processes included in this study. The humification coefficient is now updated each year according to the evolution of C input in response to soil erosion.

**2.3 Model implementation**

The soil profile has a constant thickness of 1 meter and was represented by 100 layers of 1 cm, each layer being characterized by its own clay content, SOC content, C input and turnover rates. This very fine representation of the vertical soil profile and advection in response to soil erosion is required due to sensitivity of the model to the vertical discretization as a coarse resolution typically results in substantial numerical dispersion and smoothing of the evolution of C fluxes between layers over time. Hence, to better account for time dependency of C fluxes evolution in response to soil erosion and changing C inputs, a very fine representation of soil layers was required. Test simulations showed that 100 layers represent a good compromise between computational efficiency and limited dispersion.

At the bottom of the profile, we assumed constant boundary conditions. Soil truncation was modelled as an upward advection of soil properties where the advection rate was proportional to the amount of soil removed by erosion at the surface. As we assumed a constant bulk density of the fine soil fraction, the amount of clay and SOC vertically transferred between layers was proportional to the amount of erosion (upward transfer). The SOC content in the profile was then updated each year in response to the vertical advection of matter, new C inputs at the surface and the clay content evolution following erosion. The model keeps track of the SOC and clay content per layer and tracks the evolution of crop productivity over time.

After a model spin-up without erosion allowing the C pools to reach equilibrium, we performed transient simulations where the soil profile was modified by erosion. During the simulations, erosion rates are assumed to be constant through time. We presented the results in terms of the total SOC content evolution for the 1 m profile and the net vertical C fluxes exchanged with the atmosphere. The annual net vertical flux of C between the soil and the atmosphere, integrated over the 100 soil layers at a time $t$ was calculated as follows:

$$Cv\,(t) = \sum_{z=1}^{100} i(z,t) - k_y\,(1-h)(r\,Y(z,t) + k_o\,r\,O(z,t)) \qquad (9)$$

Where $Cv(t)$ is the amount of carbon exchanged between the soil profile and the atmosphere at time $t$ and z is the depth of the layer (cm). Positive vertical carbon fluxes denote C fluxes from the atmosphere to the soil while negative vertical carbon fluxes represent a C emission to the atmosphere. We evaluated the cumulative vertical C fluxes by integrating the vertical carbon fluxes over the entire duration of the simulation.

This study is divided into two main parts: a model evaluation in which model parametrization and runs were based on site-specific environmental characteristics and a second part addressing the long-term effect of crop productivity decline on the SOC budget. The first part is based on the environmental data and empirical results of SOC loss presented by Van Oost et al. (2007) and assess the model performance with and without the erosion-crop productivity link against empirical results. The second part is a numerical exploration over a wide range of parameter combination of the effect of the erosion-crop productivity functional form on long-term SOC losses and cumulative vertical C fluxes compared to a situation without the erosion-crop productivity link.

**2.4 Model parametrization and calibration**

To explore a wide range of environmental conditions, we parametrized and calibrated the SOC profiles for ten study sites across Europe and the US based on the published data reported by Van Oost et al. (2007). Eight sites were located in Europe and two sites in the US. The European sites represent a broad range of soil and/or climate conditions. Belgian, English and Danish sites were located in temperate regions and mainly varied from each other by their erosion rates and soil properties:

from loamy soils with relatively high erosion rates in Belgium toward more clay-loam soils and slightly lower erosion rates in for the Danish and English sites (Table 1) (Quine and Zhang, 2002; Heckrath et al., 2005). The US sites were located in Iowa and characterized by fine-textured loamy to silty soils and a temperate continental climate (Ritchie et al., 2007). Mediterranean sites were characterized by a warm and dry climate, clayey soils, high erosion rate (except for the Greek site) and similar

cultivation periods (Table 1). Based on information reported in the original studies, we tried to estimate the local erosion-productivity relationship for each site. Clearly, the erosion-productivity relations are generally poorly constrained and we therefore use a range for parameter $B$ to represent uncertainty as follows: The Greek site was characterized by a convex relationship (see Kosmas et al., 2001). Spanish and Portuguese sites experienced intense soil thinning and presented similar environmental condition to the Greek site, although without the stringent constraint on soil depth. Hence, the $B$ parameter was

allowed to vary from 0.9 to 2.5, representing a range of linear ($0.9 < B < 1.1$) to convex evolution of crop productivity in response to soil erosion (Bakker et al., 2004; Kosmas et al., 2001; Van Oost et al., 2007). The Belgian and US sites, characterized by deep soils and continued high inputs, typically show a different response: here a linear to slightly concave relationships is realistic (e.g. Reyniers et al., 2006). Given the similar environmental conditions, we used a slightly concave relationship for the US, Belgian, Danish and UK sites ($0.8 < B < 0.95$).

Model calibration and parametrization was done only on the parameters describing the initial SOC profiles. The parametrization procedure considered the three model parameters that control the shape of the SOC depth profile: C inputs from crops ($ic$, Eq 4), the root-derived C input at depth $z$ ($\varphi(z)$, Eq 6) and the depth attenuation of C mineralization ($ur$, Eq 7). Based on the reported mean annual temperature, clay content and the observed profile reported in Van Oost et al. (2007),

we optimized the shape parameters of the SOC profiles for each of the 10 sites using an inverse modelling procedure (Dlugoss et al., 2012). It should be noted that the model parameters are only optimized for the representation of a stable, i.e. non-eroding, SOC profile and, hence, represent the initial SOC profile of each site. As only one depth-explicit SOC profile was available per site, no uncertainty range could be calculated. We optimized the model parameters by minimizing the relative root mean square error (RRMSE) between the observed and simulated SOC profile (Eq. 10).

$$RRMSE = \sqrt{\frac{1}{N} \sum_{i=1}^{N} \left(\frac{C_{i,s} - C_{i,o}}{C_{i,o}}\right)^2}, \qquad (10)$$

Where $N$ is the layer number, $C_{i,s}$ is the simulated carbon content of the layer $i$ (%) and $C_{i,o}$ is the carbon content observed at the depth of the mid-point layer $i$. $N$ varies for each site due to data sampling.

We used the RRMSE metric to parametrize the SOC profiles as it ensures that both the SOC content in the topsoil and in the subsoil (i.e. the profile shape) are accurately captured by the model. This is a crucial element, as these attributes will control both the C losses intensity and timing.

**Table 1** – Observed characteristics of the study sites used for the model evaluation. Site selection observed range of relative SOC loss and cumulative vertical fluxes are from Van Oost et al (2007).

| Location | Time since start of cultivation (yr) | Time since [137]Cs deposition (yr) | Erosion rate (mm yr[-1]) | Model erosion rate range (mm yr[-1]) | Topsoil clay content (%) | Crop productivity reduction form (*B*) |
|---|---|---|---|---|---|---|
| Belgium1 | 80 | 46 | 1.13 | 1.1 – 1.15 | 5 | 0.8 – 0.95 |
| Belgium2 | 100 | 46 | 0.97 | 0.95 – 1.05 | 6 | 0.8 – 0.95 |
| Denmark | 68 | 44 | 0.99 | 0.95 – 1.05 | 30 | 0.8 – 0.95 |
| Greece | 74 | 43 | 0.4 | 0.35 – 0.45 | 28 | 0.9 – 2.5 |
| Portugal | 66 | 42 | 1.09 | 1.05 – 1.15 | 18 | 0.8 – 1.3 |
| Spain1 | 66 | 42 | 1.68 | 1.65 – 1.7 | 45 | 0.9 – 1.1 |
| Spain2 | 66 | 42 | 1.13 | 1.1 – 1.15 | 52 | 0.9 – 1.1 |
| UK | 55 | 43 | 0.95 | 0.9 – 1.0 | 15 | 0.8 – 0.95 |
| USA | 143 | 49 | 1.14 | 1.1 – 1.17 | 15 | 0.4 – 0.95 |
| USA | 143 | 49 | 0.99 | 0.95 – 1.15 | 15 | 0.4 – 0.95 |

## 2.5 Model evaluation

This part aims at comparing the model performance with and without the erosion-productivity link in terms of SOC losses compared to results available in the literature. We performed a model evaluation based on data on SOC losses obtained by Van Oost et al. (2007) from 10 sites. In this study, data on SOC profiles at different geomorphic positions (stable, eroding, deposition), climate parameters, soil texture, period of cultivation and erosion rate was gathered. Based on a space for time substitution for c. 1400 soil profiles, and the use of fallout radionuclides as a tracer for lateral SOC loss, Van Oost et al. (2007) derived both the mean annual vertical and lateral C fluxes for the period under consideration (i.e. 1954 to +/-1995). The difference between the initial SOC content and the SOC content at the end of the simulation represents the total amount of SOC lost due to erosion. To allow for an inter-site comparison, the reported estimates were normalized by dividing the rates with the site-specific initial SOC content. These data provided the reference against which our simulation results are evaluated.

In a second step, we ran the model presented in this study for each site using the environmental parameters reported in Van Oost et al. (2007) and the site-specific model parameters obtained by inverse modelling (see section 2.4). To account for the uncertainty related to the estimation of the initial SOC profiles and site conditions, we created for each of the 10 sites, a set of 1000 scenarios for which parameters values were randomly chosen in a range around their optimal (for initial SOC status) and reported values (for site specific conditions) in Van Oost et al. (2007). Therefore, each of the site-specific parameter set combines fixed values (for temperature) and randomly generated parameters inside a prescribed range assuming a uniform

distribution: *ur* and *φ* were allowed to vary by ± 2 % around the optimal value. Erosion rate, clay content and crop productivity reduction exponent (when available in observations) were selected using the reported values for each site, with respective tolerances of ± 0.05 mm yr$^{-1}$ around the reported erosion rate and ± 2 % around the reported clay content (Table 1). We performed two sets simulations: one set including the effect of erosion on crop productivity (FB) and one without the erosion effect on productivity (CTL) to evaluate the effect of the erosion-crop productivity relationship on SOC losses (see Fig. 3 and Table 2). Finally, we confronted the obtained SOC losses obtained after our site-specific simulations with the values estimated by Van Oost et al. (2007). We evaluated performances of our model for both CTL and FB.

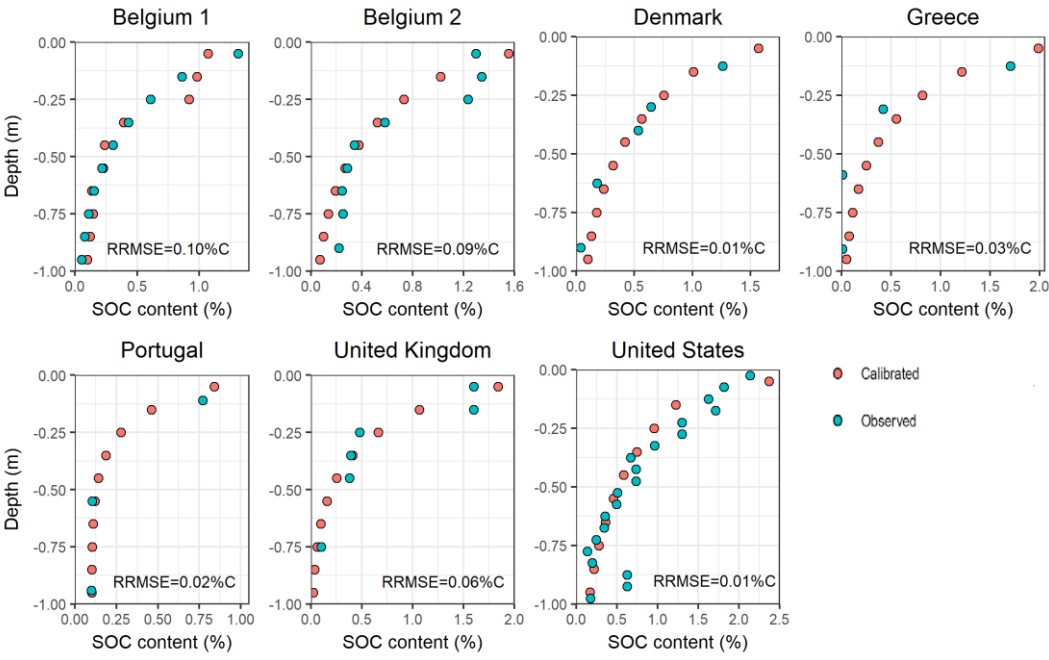

**Figure 3** – Measured (blue) and optimized (red) SOC profiles that were used to initialise the model.

**2.6 Long-term experiment**

This part aims at numerically exploring the behaviour of the model by running long-term simulations (200 years) where we focused on the effect of crop productivity, comparing the effect of the link and its shape, on the overall C budget on longer timescales. To this end, we built two sets of 1000 scenarios in which model parameters values were randomly generated, assuming a uniform distribution, in an extended range corresponding to the minimum and maximum observed values of selected sites (see Table 3). Selected parameters include the distribution with of root density (*φ(z)*, Eq 6) and the depth attenuation of C mineralization (*ur*, Eq 7), the initial clay content (*cl*), erosion rate (*E*) and crop productivity response to erosion (exponent *B*, Eq 1) vary in a larger range than the site-specific set of parameters generated previously (Table 3). We took the maximum and minimum value of each parameter reported in the site characteristic compilation of Van Oost et al. (2007) to

set the limits of the extended ranges. Erosion rates were extended up to 3 mm yr$^{-1}$, which represents a high erosion case which can be found for example in Mediterranean countries (De la Rosa et al., 2000). The root-depth parameter indicates the root penetration in the soil and its value is taken so that 95 % of the roots were distributed in the first 35 cm to 65 cm with respective values of φ of 4 to 6 (30 % to 45 % of the roots in the first 20 cm). These values are in accordance with previous SPEROS

parametrization obtained by inverse modelling (Dlugoss et al., 2012, Nadeu et al., 2015). In order to explore the effect of clay content, we linearly scaled the initial clay content profile (*cl*) creating an effective range of clay content from 5 % to 45 % in the topsoil (Table 3). We performed two analysis each one based on its own 1000 scenario set: in analysis A, the erosion rate was allowed to vary from scenario to scenario while, in the second set of 1000 scenarios (analysis B) the erosion rate was set as a constant to 1 mm yr$^{-1}$ (Table 3b). The use of these two different analysis approaches allows for an easier identification of

the role of erosion.

We performed a SOBOL procedure based on the Fourier Amplitude Sensitivity Test (FAST) to assess the contribution of each individual parameter to the global variance of the results (Cukier et al., 1973; Cukier et al., 1975). Finally, using the set of 1000 randomly-generated scenarios with variable erosion rate (analysis A), we evaluated the impact of the erosion-crop

productivity link on the SOC content and vertical fluxes after 200 years by comparing the results produced by the model in FB configuration (including the effect of erosion on productivity) and in CTL configuration (no effect of erosion on crop productivity). Note that in these long-term simulations, the reference productivity does not change as we assume constant agricultural practices. We discuss the implication of this assumption in the discussion.

**3 Results**

**3.1 Model evaluation**

In this section, we first assess the performance of the model in reproducing the observed initial SOC profiles of each site based on the calibration procedure (Fig. 3). As Figure 3 shows, the static adjustment of parameters governing the SOC profile shape for each sites resulted in a good representation of observed SOC profile. All estimated initial SOC profiles were close to the observations for each of the sites, with a RRMSE ranging between 0.01 to 0.09 (Fig. 3). In a second step, we evaluated the

model presented in this study by comparing the predicted SOC losses to those reported by Van Oost et al. (2007).

**Table 2** – RRMSE of CTL (Control dataset, no effect of erosion on crop productivity assumed) and FB (Feedback dataset, effect of erosion on productivity included) dataset for each location and as well as the RRMSE of each dataset, including all observations (all). RRSME is calculated over the whole 1m profile between observed and optimized SOC profile.


| | Relative SOC Loss | |
|---|---|---|
| Location | RRMSE CTL | RRMSE FB |
| Belgium1 | 0.18 | 0.03 |
| Belgium2 | 0.38 | 0.49 |
| Denmark | 0.09 | 0.02 |
| Greece | 0.46 | 0.43 |
| Portugal | 0.15 | 0.09 |
| Spain1 | 0.15 | 0.13 |
| Spain2 | 0.09 | 0.13 |
| UK | 0.32 | 0.37 |
| USA1 | 0.57 | 0.37 |
| USA2 | 0.52 | 0.31 |
| All | 0.33 | 0.27 |

Hereafter, the C loss will be reported as a fraction of the initial SOC content. The observed relative amount of eroded SOC at the end of the simulation varied between $0.09 \pm 0.02$ (UK) and $0.41 \pm 0.08$ (USA), with most of the values around $0.15 \pm 0.05$ of the initial content. Figure 4 clearly shows a cluster of SOC losses in this range (Table 2, Fig. 4). Site-specific simulations

produced SOC losses, which are in line with the ones estimated using the data from Van Oost et al (2007). Without the effect of erosion on productivity, the model produced relative SOC losses ranging from a $0.06 \pm 0.01$ (Greece) to $0.18 \pm 0.02$ (USA) of the initial carbon content.

Including the erosion-crop productivity relationship (scenario FB) increased SOC losses by 14 % on average and improved

the overall accuracy, albeit slightly, with an RRMSE of 0.33 for FB compared to 0.27 for CTL when all sites are were considered (Table 1). The predictive power of the model was highly site-dependent: as opposed to the results for Belgium and the USA, the Greek and Spanish sites did not exhibit a substantial increase (~ 0.5%) in SOC losses when accounting for the erosion link with productivity, relative to the CTL simulations whereas the addition of declining productivity in response to erosion increased the prediction accuracy for the environments where cumulative soil truncation was substantial (e.g. Belgium

1, Portugal, and USA). FB was able to reproduce the observed trend in which higher cumulative soil truncation leads to higher SOC losses However, the predictive power decreased the accuracy for environments with small soil truncation when the erosion-crop productivity link was taken into account (Fig. 4, Table 1). Finally, it should be noted that the margins of error of both CTL and FB modelled SOC losses are overlapping, except for the USA site.

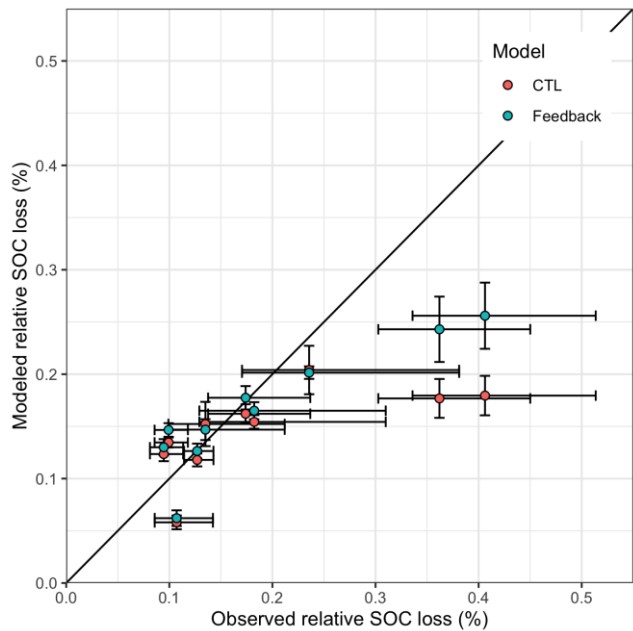

**Figure 4** – Modelled against observed relative SOC losses. Colours denote the different datasets: Control dataset (CTL, red) and the dataset including the effect of erosion on productivity (FB, blue). Error bars represent one standard deviation from the mean for both observed and modelled values.

### 3.2 Long-term simulations: sensitivity analysis

We performed a model sensitivity analysis to explore the model behaviour. The results of the FAST procedure are reported in Table 3. A sum of the contribution to the global variance may exceed 1 when two or more variable are correlated, which is the case here between erosion rate and the crop productivity response. In analysis A (i.e. erosion is included), the relative SOC loss was almost entirely controlled by (i) the soil erosion rate (70 % of the total variance) and (ii) the effect of erosion on crop productivity (18 % of the total variance) (Table 3a). Similar observations are valid for the cumulative vertical C fluxes, although vertical fluxes were more sensitive to crop productivity reduction than the erosion rate. The factor controlling the depth attenuation of C turnover was the third major factor influencing SOC losses and the cumulative C fluxes, accounting for 12 to 14 % of the variability. It should be noted that clay content and root depth distribution only played a minor role in our analysis. When the variability due to erosion was excluded from the analysis (analysis B), both SOC loss and the vertical carbon fluxes were mainly sensitive to the functional form of the link between erosion and crop productivity (~70 % of the variance) and the mineralization rate distribution with depth (23 %) (Table 3b). The root depth distribution had a weak effect on relative SOC loss only (15 % of the variance). The model simulations showed a strong positive correlation between SOC loss and erosion rate as well as with the functional form of the crop productivity reduction (colour scale, concave if B =< 0.9, linear if 0.9 < B < 1.1 and convex relationship when B > 1.1) (Fig. 5a).

**Table 3** – Selected parameters, range of tested values and results of the FAST analysis. The FAST analysis results can be interpreted as the relative contribution of each parameter variability to the total variance of the selected output, i.e. the relative SOC loss compared to the initial SOC content and the cumulative vertical C fluxes at the end of the 200 years transient simulations. Table 3a represents analysis A where erosion intensity was allowed to vary while Table 3b represents analysis B where a constant erosion rate was used. "ns" stands for "non-significant". The sum of the contribution to the global variance may exceed 1 when two or more variable are correlated.

| (a) Parameter | Symbol | Range | Relative contribution of each parameter to the relative SOC content losses variance after 200 years | Relative contribution of each parameter to the cumulative vertical C fluxes variance after 200 years |
|---|---|---|---|---|
| Erosion rate (mm yr$^{-1}$)* | $E$ | 0.1 – 3 | 0.712 | 0.201 |
| Erosion effect on productivity * | $B$ (Eq. 1) | 0.3 – 4 | 0.181 | 0.499 |
| Clay profile scaler | $Cl$ (Eq. 5) | 0.3 – 2.5 | ns | ns |
| Distribution of carbon mineralization with depth | $Ur$ (Eq. 7) | 4 – 6 | 0.139 | 0.118 |
| Root density profile with depth | $\varphi(z)$ (Eq. 6) | 1 – 3 | ns | ns |

| (b)<br>Parameter | Symbol | Range | Relative contribution of each parameter to the relative SOC content losses variance after 200 years | Relative contribution of each parameter to the cumulative vertical C fluxes variance after 200 years |
|---|---|---|---|---|
| Erosion rate (mm. yr$^{-1}$) | $E$ | 1 (fixed) | ns | ~~ns~~ |
| Erosion effect on productivity* | $B$ (Eq. 1) | 0.3 – 4 | 0.688 | 0.678 |
| Clay profile scaler | $Cl$ (Eq. 5) | 0.3 – 2.5 | ns | ns |
| Distribution of carbon mineralization with depth* | $Ur$ (Eq. 7) | 4 – 6 | 0.224 | 0.226 |
| Root density profile with depth | $\varphi(z)$ (Eq. 6) | 1 – 3 | 0.145 | ns |

### 3.3 Long-term SOC stock loss

Simulated relative SOC loss after 200 years ranged between 0.02 and 0.67 of the initial content, depending on the erosion rate
and the erosion-productivity relation used. In FB, the average SOC loss equalled 0.38 with a standard deviation of 0.13 (Fig. 5 and Table 4). When erosion rates were lower than 0.5 mm yr$^{-1}$, the simulated SOC loss was limited to 0.20 of the initial content and then increased almost linearly from 0.2 to 0.5 at an erosion rate of 1.5 mm yr$^{-1}$. Higher soil erosion rates resulted in a smaller variability in SOC losses. For example at 2 mm yr$^{-1}$, the relative SOC losses ranged from 0.32 to 0.50 of the initial SOC content. For CTL (Fig. 5b), SOC losses ranged between 0.02 and 0.57, with a mean loss of 0.34 and a standard deviation
of 0.14 (Fig. 5b and Table 4). When including the effect of erosion on productivity in our simulations (FB), SOC losses increased, particularly when using a concave and linear erosion-productivity relationship). Relative to CTL, the addition of the relationship between erosion and crop productivity further increased SOC loss by an additional 3 % to 17 % (average 7 %) after 200 simulation years (Fig. 5b and Table 4).

Although the mathematical form of the relationship between erosion and crop productivity remains uncertain, it is worth
analysing its impact over SOC losses and cumulative vertical C fluxes. Owing to its high sensitivity to soil erosion, the concave relationship (B < 0.9) resulted in the strongest relative SOC losses with an average eroded fraction of 0.43 ± 0.13 (range of 0.05 to 0.67) when compared to the results obtained in CTL (Fig. 5 and Table 4). In contrast, a linear relationship (B~1) had a weaker effect (mean eroded fraction of 0.37 ± 0.12 of the initial C stock, ranging from 0.04 to 0.61) while water availability

(B > 1.1) had the weakest effect on the mean relative SOC loss with an eroded fraction of 0.36 ± 0.12 (ranging from 0.02 to 0.62) of the initial SOC stock (Fig. 5b, Table 4).

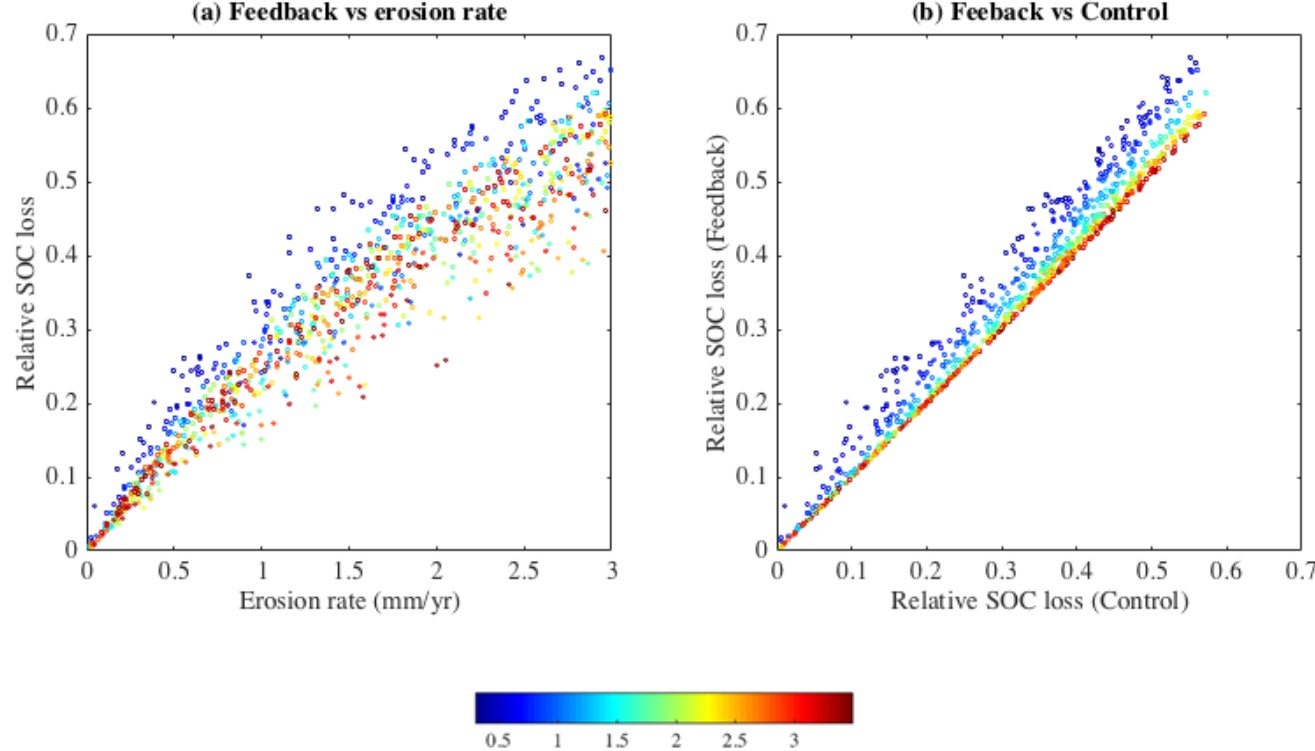

**Figure 5** – (a) Relative SOC loss after 200 years for the dataset including the effect of erosion on productivity (FB) as a function of erosion rate (mm yr$^{-1}$). (b) Relative SOC loss for FB against the relative SOC loss of CTL, for all the erosion rates. The colours scale represents the exponent $B$ value, where $B < 0.9$ a concave relationship and $B > 1.1$ represents a convex, threshold relationship. When $0.9 < B < 1.1$,

10 productivity decreases linearly with soil truncation.

### 3.4 Vertical carbon fluxes

The net cumulative carbon flux between the soil and the atmosphere after 200 years of transient simulations is represented in Figure 7. Provided that C input remained constant and unaffected by erosion (CTL), a higher erosion rate resulted in an increased net C uptake into soils due to the enhanced dynamic replacement (Fig. 6). For CTL, vertical carbon fluxes increased

15 almost linearly by 0.28 kg C m$^{-2}$ for each additional 1 mm yr$^{-1}$ of soil erosion. As expected, FB resulted in substantially lower vertical carbon fluxes (Fig. 6). However, most of the simulations still resulted in net C uptake with an average value of 0.55 kg C m$^{-2}$ ± 0.22 kg C m$^{-2}$ (- 15 % compared to CTL) (Table 4, Fig. 6, Fig. 7). While higher erosion rates generally increased the erosion-induced vertical carbon fluxes, FB induced a much larger variability, relative to CTL, particularly for the concave and linear relationships (Fig. 6).

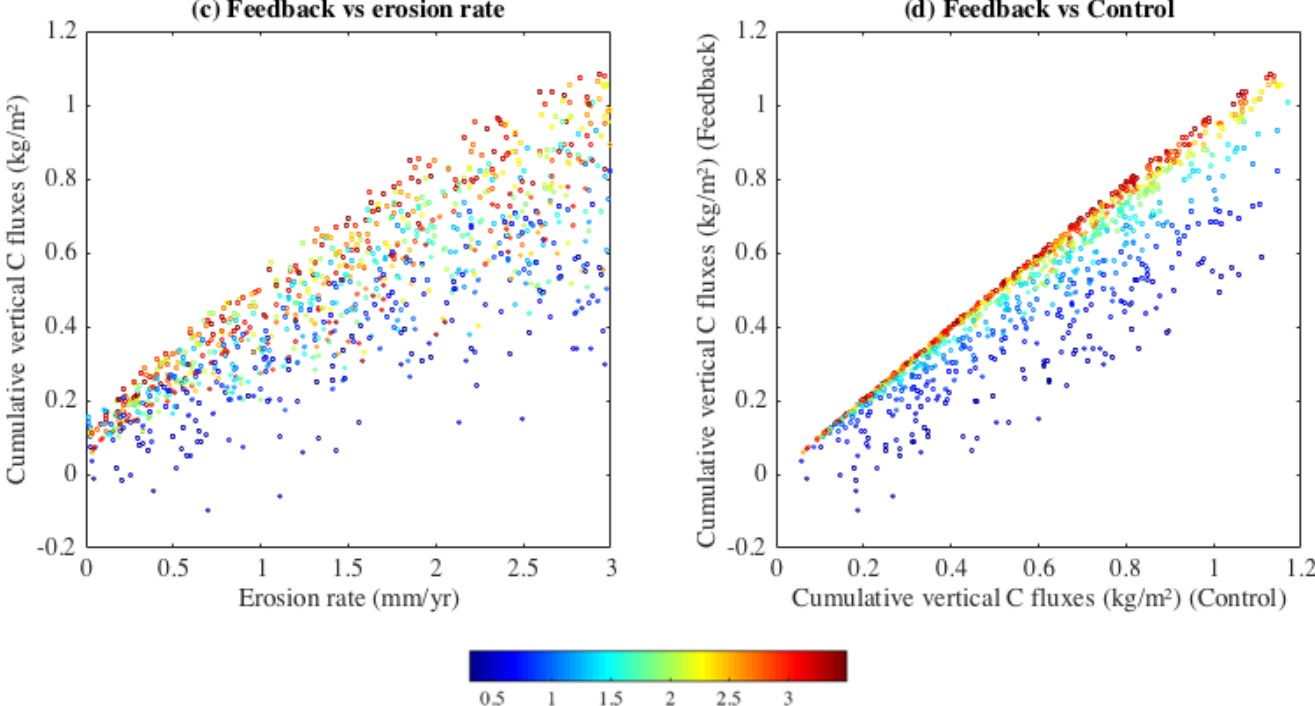

**Figure 6** – (c) Cumulative vertical C fluxes (kg C m⁻²) after 200 years for the dataset including the effect of erosion on productivity (FB) as a function of erosion rate (mm yr⁻¹). (d) Cumulative vertical C fluxes (kg C m⁻²) for FB against the cumulative vertical C fluxes (kg C m⁻²) loss of CTL, for all the erosion rates. Positive cumulative vertical fluxes represent a net uptake into soils, while negative values represent a net loss to the atmosphere. The colours scale represents the exponent $B$ value, where $B < 0.9$ a concave relationship and $B > 1.1$ represents a convex, threshold relationship. When $0.9 < B < 1.1$, productivity decreases linearly with soil truncation.

## 4 Discussion

### 4.1 Model limitations

Our study is based on several assumptions which are related to (i) the modelling framework and (ii) external factors such as agricultural practices. The first category of assumptions is mainly related to the simplifications made in linking crop productivity to C dynamics as we assumed a linear relation between C input and crop productivity. This relation may vary due to biological adaptation of plants to stress. Particularly, in shallow-soil environments or in a presence of a soil horizon limiting the growth, plants tend to adapt their root morphology and increase their root density in response to limited rooting-depth, leading to a slower decline of both C inputs and C stocks over time (Bardgett and van der Putten, 2014; Bardgett et al., 2014; Jin et al., 2017; Kosmas et al., 2001). This implies that our assumption about a constant root-depth density may result in an underestimation of soil C inputs and hence an overestimation of the C stock losses. Crop productivity reduction impacts on

SOC and vertical carbon fluxes (Figure 5) should be carefully interpreted. SOC content and cumulative vertical fluxes are much more sensitive to concave ($B < 0.9$) than threshold relationships (convex, $B > 1.1$), although this observation is a direct consequence of the nature of the mathematical function used. With only the exponent $B$ varying, the different crop productivity reduction functions used here intersect each other only when the soil truncation is zero or equals 1 meter.: under the absence of observational data, it is difficult to verify this assumption. Furthermore, the investigated soil truncation range in our simulations (60 cm) was not sufficient to surpass the threshold of crop productivity degradation when $B > 1.1$ (i.e. convex relationships). We derived the functional form of the effect of erosion on crop productivity from Bakker et al (2004) based on a limited amount of data. Although it is clear that no general and globally applicable picture emerges from this data analysis, it shows that the erosion response is not necessarily linear. The observational data nevertheless provides a range of possible outcomes, and the sensitivity analysis reported here reflects this range.

Furthermore, in our model C enrichment and preferential detachment were set to unity and we did not consider C leaching and bioturbation through the profile. The first process has been recognized as an important factor when evaluating lateral C fluxes, and particularly C export (Wilken et al., 2017). Furthermore, Gerke et al. (2016) and Herbrich et al. (2017) highlighted that environmental conditions and soil erosion had an impact of tissues growth and allocation in plants and C leaching and, hence, on the vertical distribution of C input in the profile. Our model does not take these biological parameters into account as the root-depth density is kept constant over time. Soil C leaching and bioturbation are two important factors of SOC dynamics: Gerke et al. (2016) and Herbrich et al. (2017) showed that eroded soil are more prone to C leaching than non-eroded soil due to interactions between soil truncation, biomass production, soil horizontation and the porosity characteristics. These profile characteristics affect the vertical transfer of water down the profile as well as the evapotranspiration rate resulting in C distribution and fluxes alterations. In our simulations, the vertical distribution of SOC is not affected by soil C leaching and bioturbation but the measured C fluxes ($\sim 0.005$ kg C m$^2$) resulting from C leaching under soil erosion were one order of magnitude smaller than the annual vertical C fluxes exchanged with the atmosphere reported by Van Oost et al. (2007) (Gerke et al., 2016; Herbrich et al., 2017). At the considered timescales, it is likely that C fluxes are dominated by soil redistribution while other processes play a minor role (Doetterl et al., 2016, Kirkels et al., 2014, Minasny et al., 2015).

Given the relatively large uncertainty on the simulated vertical C fluxes, it can be argued that site-specific relations are required to improve the predictive power of the model. This is particularly the case for concave relationships where our model overestimated the losses and underestimated C uptake. Even if we treated the different forms of crop productivity response to soil erosion as separate cases, these three relationships are not mutually exclusive under real conditions. Depending on soil erosion rates and soil properties, an eroding profile could experience different crop productivity responses over time: in a first phase, productivity may primarily respond to topsoil properties alteration by soil erosion. After several decades of soil erosion, soil depth limitations may exert a growing constrain on crop productivity, surpassing the initial topsoil related constraints.

Assumptions related to external factors include those made with respect to changes in agricultural practices. To build the relationship between erosion and crop productivity, we used data derived from comparative analysis of eroding soils and their stable non-eroding counterparts (same slope position) that have received the same management and external inputs rather than manipulation experiments, which ensure some real-world relevance. However, practices evolution such as mechanization and increased usage of amendments and fertilizers may compensate for the crop productivity loss as a result of continued erosion (Gregorich et al., 1998; Doetterl et al., 2016). Therefore, SOC content and crop productivity evolution may be partially decoupled whereby, without soil depth constraints, soil erosion does not substantially affect productivity. Erosion may still be an important driver for SOC losses in eroding landscapes (Meersmans et al., 2009; Bakker et al., 2007; Fenton et al., 2005). In intensively managed systems, fertilizer applications compensate for erosion-induced nutrient losses and that nutrient loss (i.e. topsoil limitation) may not be the most important effect of erosion whereas rooting space and water availability are more likely to be key issues as soil depth limitation constitutes a physical limit which could not easily be overcome by agricultural practices adaptations. On the other hand, our range of functional forms allowed for a representation of a wide variety of cases. In our simulations, we did not consider the increase in productivity that did occur during the last decades, however, it should be noted that this study focussed on the impact of erosion, relative to non-eroding conditions (e.g. Van Oost et al., 2005). Nevertheless, the simulated C loss and soil-atmosphere fluxes could be overestimated as higher C inputs allow for higher C stocks and this will reduce C losses.

## 4.2 Impact on SOC losses

Our results showed that the erosion effect on crop productivity increased SOC losses by 3 % to 17 % relative to CTL. This relative increase depends primarily on the cumulative amount of soil truncation and the functional form of the relationship between erosion and productivity. The model evaluation showed that the model predictions were close to the observations for sites that are characterized by relatively small soil truncation (i.e. short cultivation period or low erosion rates) (Fig. 4). FB resulted in an overall better prediction because it was able to predict the large relative SOC losses for the environments where intensive erosion took place. However, the addition of the erosion-induced productivity decline in the model led to contrasting results. On the one hand, SOC losses were higher for sites where productivity was more sensitive to erosion (concave or linear erosion-crop productivity relationships) and FB showed an increase in the model performance when SOC losses were important (Table 2, Fig. 4).

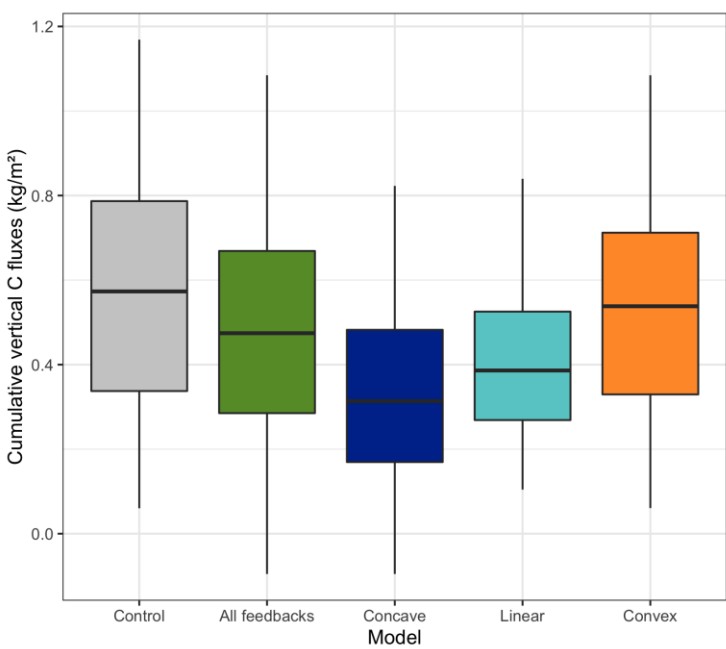

**Figure 7** – Comparison the cumulative vertical C fluxes (kg C m⁻²) between CTL (grey) and FB (colours) after 200 years of transient simulations. Positive values indicate a net flux from the atmosphere to the soil. Green, blue, light blue and orange represent respectively FB results, concave relationship, linear relationship and convex relationship. Statistics for the concave relationship, linear relationship and physical relationship are performed on subsets of FB results dataset. Boxes represent the interquartile range and whiskers the 5 % to 95 % range of the distribution.

On the other hand, linear and convex crop productivity evolution in response to soil erosion had little effect on the model results (Fig. 4, Fig. 6). The differences between FB and CTL model simulations were relatively similar except for (i) environments characterized by a concave relationship between crop productivity and soil erosion (B < 0.9) and under moderate to high erosion rates or (ii) when erosion rates are simply very high. Based on the results of the FAST analysis (Table 3), where the strong impact of cumulative soil truncation and the form of the erosion-productivity relationship were identified, we argue that the small differences between FB and CTL are mainly related to the short periods during which the sites have been exposed to agricultural erosion.

The use of longer simulation periods (200 years) further exemplified the link between erosion-crop productivity and SOC losses/vertical fluxes. The sensitivity analysis highlighted a strong influence of the soil erosion rate and crop productivity reduction rate while C profile shape, as determined by the clay content, the mineralization rate and the root depth distribution were less influential (Table 3).

**Table 4** – Relative SOC loss and cumulative C fluxes (kg C m$^{-2}$) after 200 years of transient simulations for CTL and FBs. Results are given for the whole FB and for the sub-sets of FB corresponding to the concave relationship, the linear relationship and the convex relationship.

| | Relative SOC loss | | | Cumulative vertical C fluxes (kg C m$^{-2}$) | | |
|---|---|---|---|---|---|---|
| | Range | Mean | Standard deviation | Range | Mean | Standard deviation |
| CTL | 0.01 – 0.57 | 0.34 | 0.12 | 0.06 – 1.17 | 0.59 | 0.23 |
| FB | 0.01 – 0.67 | 0.38 | 0.13 | - 0.10 – 1.09 | 0.54 | 0.22 |
| Concave | 0.05 – 0.67 | 0.43 | 0.14 | - 0.10 – 0.82 | 0.17 | 0.18 |
| Linear | 0.02 – 0.62 | 0.37 | 0.11 | 0.07 – 0.84 | 0.32 | 0.14 |
| Convex | 0.01 – 0.62 | 0.36 | 0.12 | 0.06 – 1.09 | 0.52 | 0.21 |

When the effect of erosion on productivity is not accounted for, the SOC stock follows a non-linear evolution over time that can be divided into two phases. Given the exponential form of the SOC depth profile, a quick initial decrease of the SOC content is followed by a stabilization of SOC content to a steady state level due to an equilibrium between the C input, C uptake from the atmosphere, the lateral export and the C mineralization (Bouchoms et al., 2017; Kuhn et al., 2009; Liu et al., 2003). Under continuous erosion, the rate of C export from a profile is decreasing over time owing to the differential SOC distribution between subsoil and topsoil (Kuhn et al., 2009; Liu et al., 2003). Hence, the fast initial decrease of the SOC stock is linked to the erosion of a SOC-rich topsoil, whereby a small sediment flux may carry a relatively large amount of SOC (Kirkels et al., 2014). In the later stages of the transient simulation, i.e. where the SOC-poor subsoil is exposed to erosion, the SOC loss is smaller for a similar amount of soil truncation (Kirkels et al., 2014), the impact of the erosion-crop productivity effect becomes more important and drives the SOC stock decline. Depending on the erosion rate, the first phase could last for several decades before a steady state is reached. The impact of declining productivity on the SOC losses depended on the form of the response: concave or linear responses to soil erosion tended to amplify the SOC losses in the first decades while the effect of the convex relationship may initially be partially masked and become more stringent only in the later stages of the transient simulations when compared to C losses evolution without an effect of erosion on productivity.

## 4.3 SOC dynamics in eroding landscape: discussion of the addition of erosion-crop productivity relationship on the vertical C fluxes

In eroding landscapes, several studies have highlighted that a fraction of the erosional SOC loss is replaced by new photosynthates, thereby creating a local atmospheric carbon sink (Harden et al., 1999; Berhe et al., 2007; Van Oost et al., 2007a). Although much smaller than the C release rate from land cover conversion or SOC lateral export, this erosion-induced atmospheric sink term operates on long time scales and can be sustained as long as (i) new C-depleted subsoil material is exposed to the surface and (ii) new C inputs, mainly from plants, are available (Doetterl et al., 2016; Wang et al., 2017; Naipal et al., 2018). Both conditions can be questioned here, particularly for landscapes having experienced intense cultivation, and

hence erosion, for several centuries. The first condition requires deep soils without depth limiting factors. The second condition requires continued C inputs via roots and plant residues.

In their meta-analysis, Bakker et al. (2004) highlighted that deeply truncated soils exhibit a large reduction in crop productivity. Our results showed that reducing C input in response to long-term erosion actually decreased the SOC stocks by 5 % to 67 % for the sites where intense erosion takes place (Fig. 5) and were consistent with observed SOC losses (see above and Fig. 4). As Harden et al. (1999) and Doetterl et al. (2016) reported, taking into account the erosion effect on productivity leads to a better estimation of the C budget. Hence, the dynamic replacement is likely to be overestimated when ignoring the erosion-crop productivity relationship, particularly when considering longer timescales. Our study supports these assertions: when comparing FB and CTL, the cumulative vertical C fluxes decreased on average by 15 % to 65 % after 200 years depending on the relationship nature between erosion and productivity (Fig. 6 and Fig. 7). Simulations pointed out that intense sustained erosion coupled combined with a strong reduction in soil C input can turn the soil into a net C source for the atmosphere when the soil C input becomes smaller than the mineralization rate due to decreasing productivity.

## 5. Conclusion

Based on the observations compiled in a meta-analysis, we derived a range of possible functional relationships linking soil truncation and crop productivity. We implemented the effect of soil erosion on crop productivity in a simple but depth-explicit model of SOC turnover. The integration of the erosion-productivity relationships allowed us to represent the effect of erosion on SOC evolution induced by a decrease in soil C inputs as well as from lateral SOC export. By confronting model simulations with published, and independently derived, data on lateral SOC losses and erosion-induced soil-atmosphere exchange, our results suggest that introducing erosion constraints on soil C input improves estimates of SOC losses, compared to a model approach where the effect of erosion on productivity is not included, if (i) soil truncation is substantial and (ii) the erosion-productivity relationship is accurately representing local conditions.

A sensitivity analysis showed that the erosion rate, the form of the erosion-productivity relation and the depth attenuation of the SOC mineralization rate are the key factors controlling SOC losses and soil-atmosphere C exchange. Long-term simulations showed that both SOC content and the cumulative soil-atmosphere C exchange were largely influenced by soil truncation and productivity decrease due to erosion. The inclusion of the erosion effect on crop productivity in model simulations leads to higher SOC losses (an additional SOC loss of 3 % to 17 %, relative to simulations where no link is considered) and less C uptake on eroding sites (~10% overestimation). The results are thus particularly relevant for longer-term assessments and they stress the need for an integrated landscape modelling to better constrain the overall SOC budget. Although fertilizer application may compensate for erosional nutrient losses, our simulations show that erosion-induced reduction in soil C inputs may be relevant for the soil C budget, particularly when rooting depth and water availability are limiting factors.

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
