# Peer review of "Evaluating the effects of soil erosion and productivity decline on soil carbon dynamics using a model-based approach"

_SOIL, 2018_

## Referee Comment (RC1) · Anonymous Referee #1 · 2 Aug 2018

I found this a generally well written and interesting manuscript on a timely topic. However I feel that the authors have to sharpen their arguments and they should remove several formal weaknesses (especially regarding mathematical notation and use of units), which make it difficult understanding the text.

The manuscript deals with the influence of accumulating erosion on yield and in turn on carbon sequestration. However, it ignores basic agronomic knowledge and agricultural concepts and thus has (presently) limited real-world relevance. From an agronomic point of view it is very clear that soil truncation has NO influence on yield in contrast to the basic assumption by the authors. My strong statement is easily proven because highest yields are possible without any soil (for extreme examples see the conceptual studies for a future Mars mission). What changes is not yield but the effort-yield relationship. The effort may increase to maintain yield. Some erosion effects can be changed with little or even no effort (e.g. nutrient losses in over-fertilized landscapes); other may require more effort (e.g., irrigation to compensate losses in water holding capacity). The authors may wish to argue that their relation holds true for a given and constant effort. Such behaviour may be found in controlled plot experiments but it is agronomically invalid because it would require that a farmer stops making decisions while in fact he has to decide and adjust his management every day. There is no other explanation why farmers accept soil losses that are above what soil scientist regard tolerable than that they regard the increase in effort to maintain yields smaller than the efforts needed to lower erosion. Note that usually it is assumed that erosion decreases productivity. This is something different than yield and switching from productivity to yield is not a trivial modification and would call for a discussion of its implications.

I wonder why EPIC was not used. Doesn't this do essentially the same job but allows a better control of agronomic practices and all other parameters that influence yields (which all are completely ignored in the manuscript). EPIC would allow deriving yields from productivity. This also leads to the next influence that the authors do not consider: some causes of productivity decline by soil truncation are difficult to remove while this is easy for others. For instance the authors expect the largest effect on SOC decline from a loss of nutrients due to erosion (although this is pure speculation). Such a loss of nutrients would be easy and cheap to replace in many countries. Reversibility of productivity again points to the importance of the effort-yield relationship.

The basic relation between soil depth and yield is given in figure 1. This figure suggests that the study used data but this is misleading. In fact only one conceptual relation was used although the authors suggest that this relation can be separated into three different cases. My main critique regarding this figure is twofold:

(i) It ignores a fourth rather common case, namely that productivity first increases with increasing soil truncation (often up to a truncation of 20 cm to 40 cm) and then starts to decrease. This behaviour can be found in many loessial landscapes and the effect is so strong that at least in former times without subsidies farmers paid higher prices for land where the clay depleted AE horizon had been lost and the better structured Bt horizon improved the properties of the Ap.

(ii) The interpretation of these three conceptual cases is brave. The authors explain a steep decrease in yield at little truncation by nutrient limitation. This is quite opposite to text book

knowledge of plant nutrition. Since the early times of Mitscherlich we know from the law of diminishing returns that a reduction in nutrient availability has little effect when starting at high availability. For the topic of the manuscript it is completely irrelevant whether the one curve is caused by nutrient loss and the other curve is caused by loss of water holding capacity. These interpretations, which are repeatedly treated in the manuscript like truth although any proof is missing, should entirely be removed.

[Figure]

I would suggest that the authors strictly follow the rule of notation in mathematics. E.g.: sometimes they ignore the multiplication sign and AB means $A \times B$, in other cases AB means one variable; sometimes variable are in italics, sometimes not; sometimes even mathematical signs are in italics (it should be d$t$). Units are similarly ambiguous (e.g., the unit coulomb is reported but not meant). I suggest following the "Guide for the Use of the International System of Units (SI)" (https://physics.nist.gov/cuu/pdf/sp811.pdf).

The data that were used to calibrate the model come a bit out of the blue. "we used data from ten study sites" but I am not sure whether the five references distributed within this paragraph were the origin of the data. Without clear reference there is no information about their reliability and the boundary conditions under which they were carried out.

Some assumptions inherent in the model and some equations seem doubtful and would need better justification or modification:

(i) The model treats organic manure and plant residues identical (eqn 2). I wonder whether this is true because digestibility of fresh plant material is around 75%. Hence only 25% is left after the passage of the digestive tract and it is likely to assume that he remaining 25% are more resistant to further degradation than the initial material. Furthermore, in solid manure often stabilization processes take place that do not occur with plant residues on the field.

(ii) Surprisingly, the humification factor then distinguishes between manure and crop residues although this is not possible at this stage anymore because eqn2 has already mixed manure, crop residues and other young carbon into one young pool.

(iii) The model considers only temperature as climate and edaphic (!?) factor (which temperature is not said), while usually soil moisture is the most dominant influence on SOC stabilisation (see Jenny 1941).

(iv) The model does not consider any preferential loss of SOC or clay by erosion. The results may thus only be valid for tillage erosion

(v) Only roots incorporate SOC into subsoil. Bioturbation, leaching and other processes are omitted.

(vi) It is not clear, what follows in the model below 100 cm depth. I had the impression hard rock (i.e. the model does not shift the entire soil profile downward, when topsoil is lost). In this case, the model would be far too simple because hydrology then becomes tricky. Lateral water movement could not be ignored anymore when large parts of the soil were removed. Modelling would be easier and the results likely more realistic if soft rock would follow below.

(vii) Eqn (9) seems to be wrong because all carbon that leaves the young pool is delivered to the atmosphere although large part of this carbon (see humification factor) enters the old pool.

(viii) A value of 0.55 or 0.6 seems to be more appropriate for alpha than 0.7. This could have considerable influence on the results.

The authors decided to use RRMSE for optimization (eqn (10)). Why? Isn't this a bad decision because it puts larger weight on layers with low SOC content although those layers are rather unimportant and relative measuring error is larger there? The authors also seem to have forgotten that they used RRMSE because they frequently report units of RRMSE (e.g. in Fig. 2) although this parameter cannot have a unit.

Is a model error of 93% or even 121% acceptable (see Table 1b)? I would not be satisfied.

Table 2a+b: How can the contribution of all parameters sum up to more than 100%?

Table 2b: How can erosion rate have an influence on the result although erosion rate was set constant?

The Results chapter does not differ in style and content from the preceding chapters, which were assigned to Material and Methods. Most results are in fact reported in the preceding chapters. The manuscript requires better structuring

Fig. 4: Units of the left panel? Shouldn't be a time unit in the right panel? What do the black lines denote?

Fig. 5+6 are in poor quality. Use the same font size as in fig. 4

Fig. 7: the information about the treatments is repeated three times (twice in the figure and once in the caption). What do the boxes and whiskers show (there is no convention on this)?

I didn't like the Discussion. What I missed at the very beginning is a paragraph about the assumptions and simplifications of the model and which influence they can have on the results (a little bit on this can be found at the very end but this is not stringent enough). Be more critical regarding your work. This would increase its value. At the moment it is of little value for me because I do not know under which conditions the results would apply and under

which conditions nothing could be said. Studies are cited which seem to be in agreement with your results but this does not mean much. It only becomes meaningful if we know your assumptions and simplifications because then we also know that these assumptions and simplifications would not be important for the other study.

On the other hand there are parts in the discussion that could be written even without the preceding results (e.g. the last paragraph of chapter 4.1). They could be deleted in order not to increase the length of the discussion. Also all speculations about hindrance or nutrients should be deleted. They are all unsubstantiated and misleading.

Details:

In general, the use of blanks is strange. After semicolon the authors do not like blanks. Also periods are often omitted (e.g. in i.e.)

Figure 2 only allows for yield reduction. Yield increase would also be possible (as in the already mentioned case of alfisols or in the case where an acidified topsoil is lost; there may be more cases).

I wonder why the authors used different orientation of Table 1a and 1b. The same orientation in both parts would be possible. I suggest using the same orientation as in Table 1b also in Table 1a because this is the standard orientation (variables in columns, cases in rows). Table 1 b shows the vertical C balance. In all other cases this is called vertical C flux (at least I assume that this is the same). Be consistent.

Fig. 3: Aren't the red profiles calibrated profiles (the word "simulated" would then be misleading). I thought the manuscript was about arable soils but apparently these soils do not have a plough horizon. Is the manuscript about grassland or woodland soils?

The manuscript frequently reports 1000 parameters. Fortunately the model has less. I guess the authors mean 1000 parameter sets.

There are many more technical details (e.g. inconsistent tenses, omitted periods and blanks, inconsistent formatting of references) but given that large changes are necessary it does not make sense reporting these details.

---

## Referee Comment (RC2) · Anonymous Referee #2 · 10 Sep 2018

GENERAL COMMENTS

The interdependencies of erosion and soil carbon balance have been investigated in many model-based studies. In a next step, vegetation should be explicitly included in integrated simulation approaches. Therefore, the authors tackle a relevant topic.

The general approach of the study is suitable to investigate the interdependency of erosion, plant growth and soil carbon balance. Nevertheless, the implications of the chosen implementation are not clearly addressed and are not sufficiently considered in the interpretation of the results. The study contains several flaws, which need to be addressed by the authors to make the results publishable in SOIL (see below).

[Figure]

In addition, the manuscript is not carefully prepared, hard to read, and hard to understand. This is mainly due to the lack of a common thread and to the fact that the authors use different terms for the same thing throughout the manuscript (e.g. net flux and cumulative flux and vertical flux, erosion and soil truncation, etc.). The mathematical notation is not clear and does not follow a general concept. Therefore, the text requires a complete revision to make it suitable for a scientific journal.

SPECIFIC COMMENTS

In the following paragraphs, I address the main problems I found concerning methodology and presentation of the study:

The study applies a very simple SOC model together with an equation that relates yield to erosion and an approach to translate this into depth-dependent carbon input to the soil. From my point of view, this model must not be labeled "integrated", since that would require a plant growth model. Therefore, the title of the paper should be changed. The same applies to the statement the model would dynamically link crop yields, soil properties and SOC dynamics. The model does not contain a dynamic link from soil properties to crop yields but a static assumption on the effect of erosion.

The authors use two scenarios, one of which they call FB (feedback). However, Fig. 2 and the model description reveal, that actually there is no feedback loop in the model. Using the term "feedback" is therefore misleading. I suggest using a term like "yield effect".

A central point of the study is that agricultural yield changes with soil truncation. However, there is no direct link between these variables. Soil carbon input depends more on total biomass than on yields. However, the fraction of the harvested plant organs from the total biomass (harvest index) is physiologically controlled and therefore it is variable. As the authors point out, there can be different causes for the effect of erosion on plant growth. In the real world, farmers take measures to compensate for these

effects. These simplifications need to be addressed when describing the general approach of the study and have to be included in the discussion. In this context, objective iii where the authors state their intention to investigate long-term effects of erosion on crop growth also needs to be rewritten.

In the results, the authors present data on relative yield. Here, an explanation on how the reference value was set by Bakker et al. (2004) is missing. This is crucial in order to assess the results.

The model description is hard to understand because different terms are used for the same thing (e.g. input from crops vs. flux from the atmosphere). It requires a more precise presentation. In addition, the following points have to be addressed:
- the timestep of the model has to be given
- equation 2 and 3: If $h \neq 1$, where does $(h-1)k_y rY$ go? This is only implicitly stated in Eq. 9
- using 100 soil layers seems very detailed compared to the very general assumptions on vegetation effects and C input from roots. Why did the authors choose 1 cm for layer thickness?
- Eq. 9: it should be stated, that this is just the sum of equations 2 and 3. One could factor out r, which would also simplify Eq. 3
- values for $\delta$, $k_{yt0}$, and $k_{ot0}$ are missing
- Eq. 4 contains manure input, however there is no further information on this
It also remains unclear, how soil truncation is modelled. Are layers removed from the top? Are the properties of the existing layers altered while keeping the overall soil depth constant? This has to be presented (considering the proposed effect of soil depth on plant growth).

The next point concerns the model validation. As far as I understood the text, the same observational data was used for validation and calibration. If I got this wrong, clarifying text has to be added. If I am right, this is not a validation but an evaluation. Nevertheless, a validation is required and can be accomplished by using a leave-one-out or

bootstrapping approach. As I also commented in the context of the long-term experiment, the validation needs to be conducted with the same set of perturbed parameters as the following experiments. The text states that Fig 3 shows a comparison of simulated SOC content and observations. This comparison should also include uncertainty information resulting from the 1000 simulation runs with perturbed parameters.

After the model validation, the reader will be interested in results of the model runs. How does SOC and C-exchange with the atmosphere develop over time? The authors should present timeseries that enable the reader to get an idea of how the model works. If the data were available, a comparison to observations would be desirable.

Concerning the long-term experiment, it remains unclear, why a second set of perturbed parameters was generated. In order to evaluate the results, the experiments have to be conducted with the validated model and the same sets of parameter values. In addition, information on the scientific basis of the choice of value ranges for the parameters is missing. This is of great importance if the intention of the FAST analysis is to compare the tested parameters regarding their influence on the overall variability. This is because the value ranges used for the parameters have an effect on the resulting explained total variance. In order to interpret the FAST results in the way the authors do, it has to be argued why the value ranges are comparable. Using the same relative ranges is not appropriate due to different relative ranges of the respective parameters in the field. An appropriate method is to use published ranges of observed values together with estimates of uncertainty. If these are not available, reasons for the estimates of plausible ranges have to be given.

It also is not clear to me, which set of model runs was used for the analysis in sections 3.3 and 3.4. Is this based on the same results as the FAST analysis?

Finally, the study requires a comprehensive discussion on the transferability of the results to the real world. Especially the implications of the simplifications in the model on the transferability have to be dealt with. In addition, the authors should discuss the

role of the farmers adjusting their choice of crops, management practices and harvest residuals, etc. This is tackled shorty in the final sentences of the discussion, but this is not sufficient. Other important points to be discussed are the dependency of yield on plant growth, on nutrient availability, and on access to water. All this can alter the harvest index and therefore the relation of soil carbon input and yield.

In the beginning of the discussion, results are compared to Berhe at al. (2005), which, in contrast to the present study, found a carbon sink. Explanation is required why this is rated as a support of the new results.

In the final paragraph, the authors reveal, that with B>1.1 there was no effect on yield. If this is the case throughout the study, the manuscript can be simplified by stating this in the beginning and removing this aspect in the results section.

DETAILED COMMENTS AND TECHNICAL CORRECTIONS

Use the same font and italics for symbols in equations and text unless there is an explicit rule given by the journal.

Use scientific notation for units

Improve the graphical quality of the figures.

In the following I use p for page and l for line:
p1l14: point (i) accounting for something does not change SOC fluxes but the estimates thereof
p1l22: why negative numbers for an increase in SOC losses?
p2l4f: soil-atmosphere exchanges are part of the carbon cycle. They are not its drivers, which are, by definition, external. In addition, do not exclude vegetation. Its importance is explained in the second partf of the sentence.
P2l9: incorrect format of references.
P2l23: typo: photosynthate

p2l33: no enumeration beginning here, remove (i)

p3l16: numerus: meta-analyses; experiments

p3l18f: Bakker is not cited correctly; compare to p3l27f

p3l14: this is a meta-analysis, not an analysis of meta-data

p4l3-4: unclear why a clay-fraction can replace exlpicit accounting for soil depth

p5l1: there are two van Oost 2005 papers in the references. Please specify. The same applies to some references to van Oost et al. (2007)

P5l26, Eq. 7: $K$ has to be lower case since rates were introduced lower case in Eqs. 2 and 3. In Eqs. 4, 6, and 9 dependency on time and/or layer is denoted by $t$ and $z$ in parentheses. Therefore: $k(t, z)$. In addition: explain to the reader that this is used for $k_y$ and $k_o$

p5l27: Sentence incorrect

p5l30: refer to equation 4

p5l31: to make it easier for the reader to understand the overall model setup, state the source of the cumulative soil truncation data

p7l9: two instead of 2

p7l13: remove second full stop

Table 1 a: What does "period of cultivation" mean? A single number does not define a period.

Table 1 a: table gives an erosion while caption states SOC loss. Avoid this contradiction.

Table 1 a: the caption mentions data for two simulated scenarios, which cannot be identified in the table. In addition, site description and results should not be in the same table

Table 1: scenarios are mentioned in the caption before they are mentioned in the text. In addition, when they are introduced in the text, they are not called scenarios.

p8l2: parameter sets (not parameters); the same at several positions later in the text

p8l7: scenarios (instead of abbreviations).

Figure 3: Again the terms: in the figure it is calibrated and observed, in the caption it

is measured and simulated

p10l2: where do the years come from? Were these the same for each site? Is this somehow connected to the periods in table 1?

p10l3: what is the "period of interest"?

P10l5: the parameter sets were not obtained by calibration. This only applies to the mean values.

P10l10: You investigate the feedback effect in the model. This is not a potential effect. Only transferring it to the real world makes it potential.

P10l10: what does the "c." mean?

Table 2: use the same symbols as in the text; $\varphi$ was introduced as a carbon input profile, not a root density profile. This also applies to p13l13 and p13l16

p10l16-18: sentence unclear

p11l11: instead of "typical values", state how the numbers were computed

Figure 4: consequently use upper or lower case letters to address the graphs of the figure

p12l1: the highest observed SOC loss is said to be 0.19. However, in Fig 4 a, the highest red circle is slightly above 0.2.

Figure 5: If the same variable is on both y-axes, the axis labels have to be the same.

Figures 5 and 6: When comparing simulation and observation or results from different scenarios, the graph should be square.

P19l5: Bouchoms et al. (2017) missing in list of references -> use a reference managing software to avoid this

P19l6: this is a nice explanation of the possible interaction of processes. The authors should consider presenting this in the introduction.

P19l20f: sentence unclear

P22l30: Typo: Impact of

---

## Author Comment (AC1) · 22 Nov 2018

**Reply to the reviewers**

Below, we have copied the relevant sections of the two reviews (copied sections are given in black, italic font). We have addressed the reviewers' comments separately, and have given our reply below each copied 'reviewer' section (reply is given in blue font).

**Anonymous referee #1**

I found this a generally well written and interesting manuscript on a timely topic.

Thank you for this positive assessment

However I feel that the authors have to sharpen their arguments and they should remove several formal weaknesses (especially regarding mathematical notation and use of units), which make it difficult understanding the text.

The manuscript deals with the influence of accumulating erosion on yield and in turn on carbon sequestration. However, it ignores basic agronomic knowledge and agricultural concepts and thus has (presently) limited real-world relevance. From an agronomic point of view it is very clear that soil truncation has NO influence on yield in contrast to the basic assumption by the authors. My strong statement is easily proven because highest yields are possible without any soil (for extreme examples see the conceptual studies for a future Mars mission). What changes is not yield but the effort-yield relationship. The effort may increase to maintain yield. Some erosion effects can be changed with little or even no effort (e.g. nutrient losses in over-fertilized landscapes); other may require more effort (e.g., irrigation to compensate losses in water holding capacity). The authors may wish to argue that their relation holds true for a given and constant effort. Such behaviour may be found in controlled plot experiments but it is agronomically invalid because it would require that a farmer stops making decisions while in fact he has to decide and adjust his management every day. There is no other explanation why farmers accept soil losses that are above what soil scientist regard tolerable than that they regard the increase in effort to maintain yields smaller than the efforts needed to lower erosion. Note that usually it is assumed that erosion decreases productivity. This is something different than yield and switching from productivity to yield is not a trivial modification and would call for a discussion of its implications.

We welcome your comment and clarify that our study is based on observed relations between biomass productivity and soil erosion. We would like to emphasize that the data used to construct the functional relationships are not derived from manipulation experiments but from the comparative analysis of eroding soils and their stable non-eroding counterparts that have received the same external inputs. They thus represent typical farming management and we therefore argue that they are representative and have some real-world relevance.

We agree that farmers in high-input systems will take measures to compensate the loss in crop yields. Heavy mechanization and intensive practices have contributed to increase yields and cope with most of the otherwise expected decline. However, as several studies pointed out (see Fenton et al. 2005, Reyniers et al, 2006, Kosmas et al., 2001), these measures may not be sufficient in low to medium input production systems and may not fully compensate the decline in biomass productivity. Also, changes in management practices can take time, and there might be a time lag between the increase in external inputs and the decrease in productivity, resulting in declining crop yields.

Finally, even in high-input system, it has been shown that the spatial patterns of biomass production are related to topography-driven erosion processes (e.g. Reyniers et al 2006).

The presented data do not result from an assumption about the relationship between soil erosion and biomass productivity but from observed cases in eroding landscapes under controlled amendment. We propose to discuss this explicitly in the revised manuscript.

I wonder why EPIC was not used. Doesn't this do essentially the same job but allows a better control of agronomic practices and all other parameters that influence yields (which all are completely ignored in the manuscript). EPIC would allow deriving yields from productivity. This also leads to the next influence that the authors do not consider: some causes of productivity decline by soil truncation are difficult to remove while this is easy for others. For instance the authors expect the largest effect on SOC decline from a loss of nutrients due to erosion (although this is pure speculation). Such a loss of nutrients would be easy and cheap to replace in many countries. Reversibility of productivity again points to the importance of the effort-yield relationship.

The main goal of the paper was to assess the potential impact of soil erosion on crop productivity and yield, assuming no changes in external inputs. Although we agree that it would be interesting to include the effects of agricultural management practices in the model, this is beyond the current scope of our study as we do not have the necessary input data to constrain the spatio-temporal evolution of external inputs. We will revise the literature review on mechanistic SOC models, and further clarify the scope of our paper in the introduction.

In our study, we used the relationship between soil truncation (as a result of soil erosion) and relative yields published by Bakker et al. (2004) in a process-based SOC dynamics model. The simple model structure allows us to keep the number of input parameters in balance with the available data input.

The basic relation between soil depth and yield is given in figure 1. This figure suggests that the study used data but this is misleading. In fact only one conceptual relation was used although the authors suggest that this relation can be separated into three different cases. My main critique regarding this figure is twofold:

(i) It ignores a fourth rather common case, namely that productivity first increases with increasing soil truncation (often up to a truncation of 20 cm to 40 cm) and then starts to decrease. This behaviour can be found in many loessial landscapes and the effect is so strong that at least in former times without subsidies farmers paid higher prices for land where the clay depleted AE horizon had been lost and the better structured Bt horizon improved the properties of the Ap.

Our Figure 1 is based on data that were published in the review paper by Bakker et al. (2004) on "The crop productivity-erosion relationship: an analysis based on experimental work". This publication compiled data from 24 experimental studies, and they analyzed the effect of soil truncation on yield by comparing the yield to a reference yield. Following their review, we used a subset of these data that exemplify the relationship between soil truncation and yield. In the dataset published by Bakker et al (2004), there is one case study (Olson et al 1999) where an increase in yield was observed as a result of soil truncation. This change was reported to be maximum 1.1 times the reference yield, and was observed after 25cm of soil truncation:

The Olsen et al (1999) study shows a decline in relative yield to 0.9 for the first 7.5 cm of soil truncation, followed by a slight increase to 1.1 relative yield. Furthermore, in the Belgian loess belt, Reynier et al (2006) studied the effect of soil truncation on yield and found that, even with soil amendment, yields were lower on the slopes than on the plateau as a result of soil truncation and removal of topsoil. Fenton et al. (2005), and Gregorich et al (1998) came to similar conclusions for sites in the US, and Dusar et al. (2011) and Kosmas et al. (2001) for the Mediterranean Region. The latter two studies indicated that, even with input of fertilizers, yields were decreasing because of soil erosion.

We agree with the reviewer that yield increases are possible for specific cases but the available data suggests that it may not be representative for a more generally applicable soil erosion – productivity relationship.

(ii) The interpretation of these three conceptual cases is brave. The authors explain a steep decrease in yield at little truncation by nutrient limitation. This is quite opposite to text book knowledge of plant nutrition. Since the early times of Mitscherlich we know from the law of diminishing returns that a reduction in nutrient availability has little effect when starting at high availability. For the topic of the manuscript it is completely irrelevant whether the one curve is caused by nutrient loss and the other curve is caused by loss of water holding capacity. These interpretations, which are repeatedly treated in the manuscript like truth although any proof is missing, should entirely be removed.

Based on the experimental data published in the meta-analysis by Bakker et al (2004), we have identified one mathematical expression that allows to express the change in relative yield as a result of soil truncation. The use of a simple mathematical expression facilitates its integration in the SOC dynamics model. Bakker et al. (2004) state that, following the literature, the three main regressors explaining yield losses due to soil truncation are water-availability, nutrient depletion and physical hindrance.

We agree with the reviewer that the interpretation of the three functional forms is not always straightforward. As this is not the main point of the paper, we will rework this section and remove the interpretations.

I would suggest that the authors strictly follow the rule of notation in mathematics. E.g.: sometimes they ignore the multiplication sign and AB means A × B, in other cases AB means one variable; sometimes variable are in italics, sometimes not; sometimes even mathematical signs are in italics (it should be dt). Units are similarly ambiguous (e.g., the unit coulomb is reported but not meant). I suggest following the "Guide for the Use of the International System of Units (SI)" (https://physics.nist.gov/cuu/pdf/sp811.pdf).

We apologize, and will make the necessary corrections to the annotations.

The data that were used to calibrate the model come a bit out of the blue. "we used data from ten study sites" but I am not sure whether the five references distributed within this paragraph were the origin of the data. Without clear reference there is no information about their reliability and the boundary conditions under which they were carried out.

Our apologies for this confusion. In fact, the data that were used to calibrate the model were presented in Van Oost et al. (2007), and the characteristics of each site are resumed in the supplementary material of Van Oost et al (2007). To avoid redundancy, we have referred to Van Oost et al. (2007). We will rework the text, and clarify the source of the datasets.

Some assumptions inherent in the model and some equations seem doubtful and would need better justification or modification:

(i) The model treats organic manure and plant residues identical (eqn 2). I wonder whether this is true because digestibility of fresh plant material is around 75%. Hence only 25% is left after the passage of the digestive tract and it is likely to assume that he remaining 25% are more resistant to further degradation than the initial material. Furthermore, in solid manure often stabilization processes take place that do not occur with plant residues on the field.

Manure and residues have a different humification coefficient values (respectively 0.3 and 0.125) which, in effect, leaves different amounts of C entering the first layer of the soil profile. These values result from parameter calibrations presented in the original ICBM model paper by Andren and Katterer (1997).

(ii) Surprisingly, the humification factor then distinguishes between manure and crop residues although this is not possible at this stage anymore because eqn2 has already mixed manure, crop residues and other young carbon into one young pool.

At each time step, the humification values are calculated based on the input from crop and manure at the considered time step, then the values of the C pools are computed. For the sake of clarity, the order in which the equations are presented in the text differs from the order of the calculations in the model. This probably caused the confusion and we will clarify this in the revised manuscript.

 (iii) The model considers only temperature as climate and edaphic (!?) factor (which temperature is not said), while usually soil moisture is the most dominant influence on SOC stabilisation (see Jenny 1941).

We argue that soil temperature is important for the C mineralization rate and the evolution of the SOC stocks. The ICBM model takes the moisture into account in the factor "r" (climatic factor) (Andren and Katterer, 1997)

(iv) The model does not consider any preferential loss of SOC or clay by erosion. The results may thus only be valid for tillage erosion.

We agree with the comment that we did not include selective erosion. However, based on several studies (e.g.Wang et al., 2010, 4 years monitoring), the observed enrichment is relatively small (1.3) which indicates that most of the erosion occurs under aggregated form, at least in loessical landscape. We will develop this issue in the discussion.

(v) Only roots incorporate SOC into subsoil. Bioturbation, leaching and other processes are omitted.

It is correct that our model does not take these processes into account. We argue that at a timescale of 60 to 200 years, SOC dynamics are largely dominated by soil redistribution processes and that bioturbation and leaching, although important processes on long timescales, account for a minor part of SOC fluxes and dynamics in this context (Doetterl et al., 2016, Minasny et al., 2015, Kirkels et al., 2014). We will identify this shortcoming in a revised version of the manuscript.

(vi) It is not clear, what follows in the model below 100 cm depth. I had the impression hard rock (i.e. the model does not shift the entire soil profile downward, when topsoil is lost). In this case, the model would be far too simple because hydrology then becomes tricky. Lateral water movement could not be ignored anymore when large parts of the soil were removed. Modelling would be easier and the results likely more realistic if soft rock would follow below.

We consider the following boundary condition: the soil properties (SOC, clay) observed at 1 m depth are representative for the soil/soft rock below 1 m. In the model implementation, the values of SOC and clay at the 100th layer are assumed to represent the soil characteristics below 1 m. Assuming a constant bulk density of the soil, soil characteristics are advected upward in response to soil erosion, proportionally to the amount of removed topsoil. As a response to soil erosion, the soil properties of the 100th layer are also continuously advected upward. In the physical hindrance case, the amount of coarse fragments is actually given by the low absolute clay content (in volume). We will describe this boundary condition more clearly in the revised manuscript.

(vii) Eqn (9) seems to be wrong because all carbon that leaves the young pool is delivered to the atmosphere although large part of this carbon (see humification factor) enters the old pool.

Equation 9 is just the difference between the input of carbon entering the soil and the mineralized carbon leaving the young and old pools. Eq 9 is based on Fig. 1 from Andren and Katterer (1997) who developed the ICBM model. We checked Eq 9 with the original ICBM formulation (Andren and Katterer, 1997) and it is correctly presented in our manuscript.

(viii) A value of 0.55 or 0.6 seems to be more appropriate for alpha than 0.7. This could have considerable influence on the results.

The authors decided to use RRMSE for optimization (eqn (10)). Why? Isn't this a bad decision because it puts larger weight on layers with low SOC content although those layers are rather unimportant and relative measuring error is larger there? The authors also seem to have forgotten that they used RRMSE because they frequently report units of RRMSE (e.g. in Fig. 2) although this parameter cannot have a unit.

We argue that the shape of the SOC profile is as important as topsoil SOC content for our study. As Kirkels et al. (2014) pointed out, SOC stock and lateral fluxes follow a two-phase evolution in which the very high rate of loss in the first decades is followed by a period of lower loss rates. The evolution is similar for the vertical C fluxes as the C uptake increases fast at the beginning of the erosion period while the rate of increase is slowing down over time. This temporal evolution is due to the lower SOC content in the subsoil. Hence, the shape of the SOC profile determines the intensity and evolution of both lateral and vertical C exchanges. As these

fluxes are a key part of our analysis, the RRMSE was kept so that parametrization of SOC profile would ensure a good representation of (i) observed SOC profile and (ii) an accurate representation of the impact of soil erosion on C fluxes.

Is a model error of 93% or even 121% acceptable (see Table 1b)? I would not be satisfied. Table 2a+b: How can the contribution of all parameters sum up to more than 100%?

The model error is indeed high for the cumulative vertical C fluxes, in contrast to the SOC stock loss' prediction. We point out in the discussion that this discrepancy as well as the high error is mainly due to the yield effect. We hypothesize that a site-specific yield effect would allow to decrease the error, but these data were not available. Finally, the long timescales considered should be taken into account when analyzing the model errors: the model predictions are in the correct order of magnitude and the relative differences between the sites are well represented.

In FAST analysis, the sum of contributions can be more than 1 when two (or more) variables are correlated. In our case, erosion rate and yield response to soil truncation are correlated.

Table 2b: How can erosion rate have an influence on the result although erosion rate was set constant?

A FAST analysis can show small positive contributions for constant parameters when (i) the number of runs is too small and (ii) due to mathematical dispersion. In our case, both cases are plausible. This also applies to negative contributions. We will discuss this in the revised manuscript.

The Results chapter does not differ in style and content from the preceding chapters, which were assigned to Material and Methods. Most results are in fact reported in the preceding chapters. The manuscript requires better structuring

We will better separate materials and methods from results.

Fig. 4: Units of the left panel? Shouldn't be a time unit in the right panel? What do the black lines denote?

The left panel represents the relative SOC loss $1 - \frac{\text{SOC (final)}}{\text{SOC (initial)}}$. It is thus without dimension.

Fig. 5+6 are in poor quality. Use the same font size as in fig. 4

This may be due to the compression applied when submitting the manuscript. We will provide figures with better quality.

Fig. 7: the information about the treatments is repeated three times (twice in the figure and once in the caption). What do the boxes and whiskers show (there is no convention on this)?

The boxes represent the interquartile range and whiskers represent the 95 % quantiles of the distribution. We will correct the information of Figure 7 and add a description of the meaning of the boxes and whiskers.

I didn't like the Discussion. What I missed at the very beginning is a paragraph about the assumptions and simplifications of the model and which influence they can have on the results (a little bit on this can be found at the very end but this is not stringent enough). Be more critical regarding your work. This would increase its value. At the moment it is of little value for me because I do not know under which conditions the results would apply and under which conditions nothing could be said. Studies are cited which seem to be in agreement with your results but this does not mean much. It only becomes meaningful if we know your assumptions and simplifications because then we also know that these assumptions and simplifications would not be important for the other study.

On the other hand there are parts in the discussion that could be written even without the preceding results (e.g. the last paragraph of chapter 4.1). They could be deleted in order not to increase the length of the discussion. Also all speculations about hindrance or nutrients should be deleted. They are all unsubstantiated and misleading.

We agree with this comment and will add a paragraph on model simplifications and assumptions in the revised manuscript. We will also remove some of the less relevant parts of the discussion and modify the discussion as suggested by the reviewer.

Details:

In general, the use of blanks is strange. After semicolon the authors do not like blanks. Also periods are often omitted (e.g. in i.e.)

We will correct the typo and punctuations issues.

Figure 2 only allows for yield reduction. Yield increase would also be possible (as in the already mentioned case of alfisols or in the case where an acidified topsoil is lost; there may be more cases).

See discussion above. However, based on your comment, we will add this effect in our simulations and will evaluate the response of the model.

I wonder why the authors used different orientation of Table 1a and 1b. The same orientation in both parts would be possible. I suggest using the same orientation as in Table 1b also in Table 1a because this is the standard orientation (variables in columns, cases in rows). Table 1 b shows the vertical C balance. In all other cases this is called vertical C flux (at least I assume that this is the same). Be consistent.

We will change the orientation of Table 1.

Fig. 3: Aren't the red profiles calibrated profiles (the word "simulated" would then be misleading). I thought the manuscript was about arable soils but apparently these soils do not have a plough horizon. Is the manuscript about grassland or woodland soils?

We agree that these are calibrated profiles. The study is only for arable lands, and we did not take tillage into account, only water erosion.

The manuscript frequently reports 1000 parameters. Fortunately the model has less. I guess the authors mean 1000 parameter sets.

We apologize for this, it is correct that we generated 1000 sets of parameters values.

There are many more technical details (e.g. inconsistent tenses, omitted periods and blanks, inconsistent formatting of references) but given that large changes are necessary it does not make sense reporting these details.

We will carefully revise the manuscript, and pay attention to the formatting of text, references and tables.

---

## Author Comment (AC2) · 22 Nov 2018

**Reply to the reviewers**

Below, we have copied the relevant sections of the two reviews (copied sections are given in black, italic font). We have addressed the reviewers' comments separately, and have given our reply below each copied 'reviewer' section (reply is given in blue font).

**Anonymous Referee #2**

GENERAL COMMENTS

The interdependencies of erosion and soil carbon balance have been investigated in many model-based studies. In a next step, vegetation should be explicitly included in integrated simulation approaches. Therefore, the authors tackle a relevant topic.

The general approach of the study is suitable to investigate the interdependency of erosion, plant growth and soil carbon balance. Nevertheless, the implications of the chosen implementation are not clearly addressed and are not sufficiently considered in the interpretation of the results. The study contains several flaws, which need to be addressed by the authors to make the results publishable in SOIL (see below).

In addition, the manuscript is not carefully prepared, hard to read, and hard to understand. This is mainly due to the lack of a common thread and to the fact that the authors use different terms for the same thing throughout the manuscript (e.g. net flux and cumulative flux and vertical flux, erosion and soil truncation, etc.). The mathematical notation is not clear and does not follow a general concept. Therefore, the text requires a complete revision to make it suitable for a scientific journal.

The necessary clarifications on terminology and mathematical notations will be made in the final manuscript.

SPECIFIC COMMENTS

In the following paragraphs, I address the main problems I found concerning methodology and presentation of the study:

The study applies a very simple SOC model together with an equation that relates yield to erosion and an approach to translate this into depth-dependent carbon input to the soil. From my point of view, this model must not be labeled "integrated", since that would require a plant growth model. Therefore, the title of the paper should be changed. The same applies to the statement the model would dynamically link crop yields, soil properties and SOC dynamics. The model does not contain a dynamic link from soil properties to crop yields but a static assumption on the effect of erosion.

The authors use two scenarios, one of which they call FB (feedback). However, Fig. 2 and the model description reveal, that actually there is no feedback loop in the model. Using the term "feedback" is therefore misleading. I suggest using a term like "yield effect".

A central point of the study is that agricultural yield changes with soil truncation. However, there is no direct link between these variables. Soil carbon input depends more on total biomass than on yields. However, the fraction of the harvested plant organs from the total biomass

(harvest index) is physiologically controlled and therefore it is variable. As the authors point out, there can be different causes for the effect of erosion on plant growth. In the real world, farmers take measures to compensate for these effects. These simplifications need to be addressed when describing the general approach of the study and have to be included in the discussion. In this context, objective iii where the authors state their intention to investigate long-term effects of erosion on crop growth also needs to be rewritten.

We welcome this assessment.

In the current version of the model, there is no explicit link between soil properties and crop yields. Our study is based on a published relationship between soil erosion (expressed as soil truncation) and relative yield and implicitly contains these effects. The soil properties, SOC dynamics and C input (derived from yields/biomass productivity) are integrated as SOC dynamics depend on clay content and C input which are influenced by soil erosion. These links are therefore explicit. We will thus reformulate carefully the title and the text so that the difference between explicit and implicit links are clear. The feedback term will be adapted to clarify that it represents the indirect effect of erosion on SOC dynamics through yield reduction.

We agree that yield and biomass production are two different concepts, which are often mixed in the literature and common language. In this paper, we are talking about biomass productivity in response to soil erosion and we agree that farming practices will try to cope with declining biomass production. We will clarify in the implementation and discussion that we are assuming constant agricultural management practices.

In the results, the authors present data on relative yield. Here, an explanation on how the reference value was set by Bakker et al. (2004) is missing. This is crucial in order to assess the results.

We selected data from comparative plots in which the original studies compared yield obtained in non-to slightly eroded soil with yield in eroded soil. Relative yields were calculated as following: relative yield is set to 1 for the non-or slightly eroded soil and fractions of that for yields on eroded soils. Hence, a relative yield of 1 indicates that there is no change in the yield, values < 1 represent yield losses and values > 1 yield gains. We will clarify this in the text and figure captions.

The model description is hard to understand because different terms are used for the same thing (e.g. input from crops vs. flux from the atmosphere). It requires a more precise presentation. In addition, the following points have to be addressed:
- the timestep of the model has to be given
- equation 2 and 3: If $h \neq 1$, where does $(h-1)k_y r Y$ go? This is only implicitly stated in Eq. 9
- using 100 soil layers seems very detailed compared to the very general assumptions on vegetation effects and C input from roots. Why did the authors choose 1 cm for layer thickness? - Eq. 9: it should be stated, that this is just the sum of equations 2 and 3. One could factor out r, which would also simplify Eq. 3
- values for $\delta$, $k_{yt0}$, and $k_{ot0}$ are missing
- Eq. 4 contains manure input, however there is no further information on this

The model time step is 1 year, we will add this information.

Equation 2 and 3: the quantity (1-h)*k*r*Y represents the mineralized/respired fraction leaving the young C pool.

We used 100 soil layers to have a very fine representation of the vertical soil profile and advection in response to soil erosion. We found that the model was sensitive to the vertical SOC profile and using a coarse resolution resulted in substantial numerical dispersion and smoothing. In addition, as the model computational performance was very good, there is no need for a low vertical resolution.

Eq. 9 is the sum of both equations. However, Eq 2 and Eq 3 are the classic way to present ICBM equations. We refer to Andren and Katterer, 1997 and SPEROS model presentations by Van Oost et al. (2005), Dlugoss et al. (2012), Nadeu et al. (2015).

Values for ky and ko are respectively 0.8 and 0.006 yr-1 and $\delta$ is 2.91 (dimensionless). These values will be added to the manuscript.

It also remains unclear, how soil truncation is modelled. Are layers removed from the top? Are the properties of the existing layers altered while keeping the overall soil depth constant? This has to be presented (considering the proposed effect of soil depth on plant growth).

Soil truncation is modelled by removing soil properties from the top of the profile. Assuming a constant bulk density, the considered depth does not change over time but the soil characteristics are advected upward in response to soil erosion. Soil properties are upward in proportion to the amount of soil removed (see e.g. Van Oost et al, 2005, Dlugoss et al., 2012, Nadeu et al. 2015). At the bottom of the profile, a constant boundary condition is assumed and its properties are progressively included in the soil profile proportionally to the amount of removed topsoil, resulting in an effective truncation of the soil profile characteristics.

The next point concerns the model validation. As far as I understood the text, the same observational data was used for validation and calibration. If I got this wrong, clarifying text has to be added. If I am right, this is not a validation but an evaluation. Nevertheless, a validation is required and can be accomplished by using a leave-one-out or bootstrapping approach. As I also commented in the context of the long-term experiment, the validation needs to be conducted with the same set of perturbed parameters as the following experiments. The text states that Fig 3 shows a comparison of simulated SOC content and observations. This comparison should also include uncertainty information resulting from the 1000 simulation runs with perturbed parameters.

We agree that the term model evaluation is more appropriate. During this evaluation, we performed site-specific simulations as SOC parametrization, clay content, erosion rate and length of the simulations were specific for each site from the Van Oost et al. (2007) paper. For each of the 10 sites, we created a set of 1000 scenarios for which parameter values were randomly chosen in a narrow range around their published values in Van Oost et al. (2007). These values, associated ranges and lengths of simulations are given in Table 1a. We will add the uncertainty in SOC profiles calibration to Figure 3.

After the model validation, the reader will be interested in results of the model runs. How does SOC and C-exchange with the atmosphere develop over time? The authors should present timeseries that enable the reader to get an idea of how the model works. If the data were available, a comparison to observations would be desirable.

The time-series resulting for our simulations are available and could easily be included in the paper. SOC stock evolution is following a classic two phase evolution: the profiles lose quickly a large amount of carbon during the first decades, and the rate of SOC loss is then decreasing over time due to the lower SOC content of the exposed subsoil. When the yield effect is weak, a steady-state is observed whereby the laterally exported SOC is replaced by new C coming from plant inputs. When the yield effect is strong, it takes a longer time to come to steady-state SOC stocks or there is no steady-state.

To our knowledge, long-term observational data on C fluxes nor SOC stock evolution are not available in literature.

Concerning the long-term experiment, it remains unclear, why a second set of perturbed parameters was generated. In order to evaluate the results, the experiments have to be conducted with the validated model and the same sets of parameter values. In addition, information on the scientific basis of the choice of value ranges for the parameters is missing. This is of great importance if the intention of the FAST analysis is to compare the tested parameters regarding their influence on the overall variability. This is because the value ranges used for the parameters have an effect on the resulting explained total variance. In order to interpret the FAST results in the way the authors do, it has to be argued why the value ranges are comparable. Using the same relative ranges is not appropriate due to different relative ranges of the respective parameters in the field. An appropriate method is to use published ranges of observed values together with estimates of uncertainty. If these are not available, reasons for the estimates of plausible ranges have to be given.

The model evaluation was done comparing the model predictions against observations using site-specific data. These data are displayed in Table 1 and Figure 3. We added a relative uncertainty range around these observations to account for natural variability and errors in measurements at the site-scale. A range of B exponent was attributed to each site, in line with each site's description of soil depth description and climate type. For each individual site, we generated 1000 sets of parameters, which values were inside the range of this specific site. We performed 1000 simulations which time length was site-specific (i.e. 1000 simulations with the parameters of Belgium 1 site, 1000 simulations with the parameters of UK site, etc.). Therefore, the resulting SOC losses and vertical C fluxes can directly be compared to the observed values as the erosion and SOC parameters were close to the observations.

The long-term experiments were performed on the total range of the observed parameter values regardless of the sites considered in the model evaluation: i.e. from the smallest value to the highest value found in the table, with the notable exception of erosion rate, which range was extended further based on erosion data across Europe and the USA. We generated 1000 sets of parameters based on this total range of values (as presented in table 2). Specifically, the range of the yield-effect exponent was chosen to cover the whole set of yield values per unit of soil truncation as extracted from Bakker et al. (2004) and this was presented in the first part of the manuscript. The root-depth parameter indicates the root penetration in the soil and its value was taken so that 95% of the roots are distributed in the first 35 cm to 65 cm with respective values of φof 4 to 6, with 30 to 45% in the first 20 cm. These values are in accordance with previous SPEROS parametrization obtained by inverse modelling (Dlugoss et al., 2012, Nadeu et al, 2015). As for the mineralization distribution, the given range indicates a turnover rate at 1 m depth of 137 to 700 years for the slow C pool which is in line with the centennial turnover rate found in deep colluvium by Wang et al. (2014) or Van Oost et al (2012).

We thus argue that the interpretation of the SOBOL/FAST analysis is valid and we will more clearly identify in the text where the ranges of the parameters come from.

It also is not clear to me, which set of model runs was used for the analysis in sections 3.3 and 3.4. Is this based on the same results as the FAST analysis?

The results for the long-term simulations (200 years) in section 3.3, 3.4 are based on a set of 1000 scenarios randomly chosen in the range of values specified in Table 2a. This set was also used in the FAST analysis.

Finally, the study requires a comprehensive discussion on the transferability of the results to the real world. Especially the implications of the simplifications in the model on the transferability have to be dealt with. In addition, the authors should discuss the role of the farmers adjusting their choice of crops, management practices and harvest residuals, etc. This is tackled shorty in the final sentences of the discussion, but this is not sufficient. Other important points to be discussed are the dependency of yield on plant growth, on nutrient availability, and on access to water. All this can alter the harvest index and therefore the relation of soil carbon input and yield.

We will add and clarify the aspect related to the model limitations and the agricultural practices adaptation in the extended discussion about the study limitations. As for the dependency between yields, nutrient availability or water availability, these aspects have been discussed by Bakker et al in their review (2004), Christinsen and McElya (1988), Lal et al. (1999) or Larson et al. (1985). For example, Diaz-Zorita et al. (1999) pointed out that in absence of water limitation, nutrient limitation could reduce yield by 40 kg.ha$^{-1}$. We consider that a more detailed analysis of the biological effects is outside the scope of this paper.

In the beginning of the discussion, results are compared to Berhe at al. (2005), which, in contrast to the present study, found a carbon sink. Explanation is required why this is rated as a support of the new results.

Berhe et al (2005) found a carbon sink related to the C uptake from the atmosphere occurring in eroding areas. Our study found that erosion can result in a carbon sink (in terms of vertical C fluxes) as the balance is often positive with C being added to the soil. Our study however emphasizes that this C uptake can be overestimated in modelling studies if the long-term evolution of the yields is omitted.

In the final paragraph, the authors reveal, that with B>1.1 there was no effect on yield. If this is the case throughout the study, the manuscript can be simplified by stating this in the beginning and removing this aspect in the results section.

We refer to our reply to reviewer#1. The main goal of our study was to explore the effect of biomass productivity decrease on SOC losses and vertical C fluxes. We acknowledge that the model is relatively simple and required assumptions about the relationship between the C input and the biomass productivity. We agree that the relationship between C input and biomass can be dependent on the amount of residues left on the field but under the absence of data, it is difficult to correctly represent this.

DETAILED COMMENTS AND TECHNICAL CORRECTIONS

We thank the reviewer for the suggestions. For the sake of clarity, we only answer individual comments relative to understanding, clarifications, and precisions. All other comments about

typo, references or re-phrasing which do not required detailed answers will be addressed in the revised manuscript.

Use the same font and italics for symbols in equations and text unless there is an explicit rule given by the journal.

The journal asks for equation symbols to be in italic when used in the text. The necessary changes will be made.

Improve the graphical quality of the figures.

We will improve the figures.

p1l22: why negative numbers for an increase in SOC losses?

This is a mistake as the numbers represent the relative C stock changes. We will correct this.

p3l14: this is a meta-analysis, not an analysis of meta-data

We agree with this comment will change it throughout the manuscript.

p4l3-4: unclear why a clay-fraction can replace explicit accounting for soil depth

The clay fraction can be given in absolute terms, i.e. the volume of clay in a given volume of substrate (soil + rock fragment) or in relative terms as the fraction of clay in the remaining space, not occupied by rock fragments. In our case, the clay fraction is accounted for in the model by the absolute volume of clay per volume of substrate. We further assumed that the relative fraction of clay in the remaining space and the bulk density of the soil are constant. Hence, the absolute amount of clay indirectly indicates how much rock fragment is contained in the substrate, which is a proxy of soil depth as 100% of rock fragment is representing the bedrock level. In the case of deep soft rock, the absolute clay content shows little variation between the topsoil and the bottom of the profile. In the case of physical hindrance, the clay content is highly reduced at the bottom of the profile.

p5l1: there are two van Oost 2005 papers in the references. Please specify. The same applies to some references to van Oost et al. (2007)

We will make the changes

P5l26, Eq. 7: K has to be lower case since rates were introduced lower case in Eqs. 2 and 3. In Eqs. 4, 6, and 9 dependency on time and/or layer is denoted by t and z in parentheses. Therefore: k(t, z). In addition: explain to the reader that this is used for ky and ko p5l27: Sentence incorrect
p5l30: refer to equation 4

We will make the changes

p5l31: to make it easier for the reader to understand the overall model setup, state the source of the cumulative soil truncation data

We will add information about the link between erosion and cumulative soil truncation (which is the annual erosion rate * time, as erosion rate does not vary).

p7l9: two instead of 2
p7l13: remove second full stop

We will make the changes

Table 1 a: What does "period of cultivation"mean? A single number does not define a period.

The period of cultivation is the total duration of cultivation between the start of cultivation on the considered field and the date of the final analysis. We will correct the header to "time since start of cultivation"

Table 1 a: the caption mentions data for two simulated scenarios, which cannot be identified in the table. In addition, site description and results should not be in the same table

The caption was wrong and will be corrected according to the content of the table.

p10l2: where do the years come from? Were these the same for each site? Is this somehow connected to the periods in table 1?

In this case, these are different from the period of cultivation as the $^{137}$Cs was released in the atmosphere and deposited after the nuclear bomb testing. In the literature and following Van Oost et al. (2007), we took 1954 as the standard date of $^{137}$Cs deposition on the earth surface (Ritchie and McHenry, 1990). This date is considered to be identical for all sites. As the erosion rate derived from $^{137}$Cs tracer were valid for the period post-1954, the integration of cumulative vertical C fluxes was done over the period from 1954 to the date of the C inventories realized in each individual site rather than over the entire period of cultivation.

p10l3: what is the "period of interest"?

This is the period of cultivation for SOC losses or the period between 1954-date of sampling for the vertical fluxes. We will clarify this in the text.

P10l5: the parameter sets were not obtained by calibration. This only applies to the mean values.

Correct, we will change this in the text

P10l10: You investigate the feedback effect in the model. This is not a potential effect. Only transferring it to the real world makes it potential.
Correct, we will adapt this

P10l10: what does the "c." mean?

c. stands for "calibrated" years.

Table 2: use the same symbols as in the text; φ was introduced as a carbon input profile, not a root density profile. This also applies to p13l13 and p13l16

We will check the use of the symbols.

p11l11: instead of "typical values", state how the numbers were computed

These numbers of SOC losses were obtained by calculations based on the data provided by Van Oost et al. (2007): stable profile SOC stock, lateral SOC fluxes, vertical SOC fluxes and erosion rate for each site. The total C losses was calculated by integrating lateral SOC fluxes, vertical SOC fluxes and calculating a mass balance to obtain the total SOC lost over the cultivation period. The observed relative SOC loss is the ratio between the total SOC loss and the observed SOC stock. We clarify the method in the manuscript.

Figure 4: consequently use upper or lower case letters to address the graphs of the figure

We will adapt this.

p12l1: the highest observed SOC loss is said to be 0.19. However, in Fig 4 a, the highest red circle is slightly above 0.2.

We apologize, this is a mistake in the text.

Figure 5: If the same variable is on both y-axes, the axis labels have to be the same. Figures 5 and 6: When comparing simulation and observation or results from different scenarios, the graph should be square.

We will make the necessary changes.

P19l5: Bouchoms et al. (2017) missing in list of references -> use a reference managing software to avoid this

We will carefully check the formatting of references, and make sure that the list is complete.

P19l6: this is a nice explanation of the possible interaction of processes. The authors should consider presenting this in the introduction.

Thanks for this suggestion.

P19l20f: sentence unclear

We will clarify the sentence which is describing the three processes involved into the C sink resulting from erosion.

---

## Author Response (AR1)

**Reply letter**

Dear Editor and Reviewers,

Below, we have copied the relevant sections of the editors and the two reviews (copied sections are given in black, italic font). We have addressed the reviewers' comments separately, and provide our reply below (reply is given in blue font). The reply to the Anonymous Referee #1 is on p.2 and the response to the Anonymous Referee #2 is given on p.10. Furthermore, a track-change version of the manuscript is provided at p. 21 of this document.

**1. Reply to the Editor**

Dear Authors,
Following the general and specific comments of the two reviewers, a major revision of your manuscript is required to reconsider your study in SOIL. Please take all the comments of the reviewers carefully into account before resubmitting your paper.
Three aspects were underlined by both reviewers, which therefore should be especially handled with care. (i) You should be clearer regarding your assumptions and should discuss the consequences of these assumptions in more detail (especially regarding yield/biomass relations; constant agricultural management etc.). (ii) Both reviewers mentioned several times that your data utilized from Bakker et al. 2004 need to be described in more detail. (iii) There is a large number of formal errors in the manuscript which need to be removed.
Following reviewer #1, I also have some general doubts regarding the three relations given in Fig. 1, which are used to explain why erosion results in a yield decline (nutrient depletion, physical hindrance and water availability). I suggest, to discuss these reasons, but do not over interpret this as the data basis seemed to be quite weak.
Best regards,
Peter Fiener

Dear Editor, thank you for the comments. We carefully revised the manuscript according to the suggestions made by you and both reviewers. You will find the answers to their questions/suggestions and where they have been integrated into the revised manuscript below.

As for the three aspects pointed out by you:

(i)     We clarified the assumptions of the model regarding agricultural practices and yield/biomass relationship on P3 L30. We further discuss the implications of our assumptions (i.e. no evolution in agricultural practices and omitting the difference between crop productivity and yields) in section 4.1 "Model limitations" (P17 L6).
(ii)    We now present a substantially improved description of Bakker's data, i.e. how they were obtained, scaled and what they represent in section 2.1 (P3 L8).
(iii)   We carefully checked the manuscript to correct the formal errors.

Finally, we agree that these interpretations of the yield decline to soil truncation are associated with substantial uncertainty. We therefore removed these interpretations from the manuscript to focus on the effect of the mathematical form of the erosion-crop productivity link on the SOC losses and vertical C fluxes.

**Reply to the reviewers**

Below, we have copied the relevant sections of the two reviews (copied sections are given in black, italic font). We have addressed the reviewers' comments separately, and provide our reply below (reply is given in blue font).

**2.1 Anonymous referee #1**

I found this a generally well written and interesting manuscript on a timely topic.

Thank you for this positive assessment

However I feel that the authors have to sharpen their arguments and they should remove several formal weaknesses (especially regarding mathematical notation and use of units), which make it difficult understanding the text.

The manuscript deals with the influence of accumulating erosion on yield and in turn on carbon sequestration. However, it ignores basic agronomic knowledge and agricultural concepts and thus has (presently) limited real-world relevance. From an agronomic point of view it is very clear that soil truncation has NO influence on yield in contrast to the basic assumption by the authors. My strong statement is easily proven because highest yields are possible without any soil (for extreme examples see the conceptual studies for a future Mars mission). What changes is not yield but the effort-yield relationship. The effort may increase to maintain yield. Some erosion effects can be changed with little or even no effort (e.g. nutrient losses in over-fertilized landscapes); other may require more effort (e.g., irrigation to compensate losses in water holding capacity). The authors may wish to argue that their relation holds true for a given and constant effort. Such behaviour may be found in controlled plot experiments but it is agronomically invalid because it would require that a farmer stops making decisions while in fact he has to decide and adjust his management every day. There is no other explanation why farmers accept soil losses that are above what soil scientist regard tolerable than that they regard the increase in effort to maintain yields smaller than the efforts needed to lower erosion. Note that usually it is assumed that erosion decreases productivity. This is something different than yield and switching from productivity to yield is not a trivial modification and would call for a discussion of its implications.

We welcome this insightful comment. We clarified in our manuscript that our study is based on observed relations between biomass productivity and soil erosion. We would like to emphasize that the data used to construct the functional relationships are not derived from manipulation experiments but from the comparative analysis of eroding soils and their stable non-eroding counterparts (same slope position) that have received the same management and external inputs. They represent actual farm management practices and we therefore argue that they are representative and have some real-world relevance. We have worded this more carefully in the revised manuscript (P3 L30).

We agree that farmers in high-input systems will take measures to compensate the loss in crop yields. Agricultural intensification has resulted in increased yields and this has masked the expected decline related to erosion. However, as several studies pointed out (see Fenton et al. 2005, Reyniers et al, 2006, Kosmas et al., 2001), these measures may not be sufficient in low to medium input production systems and may not fully compensate the decline in productivity, particularly when nutrient losses are not the main cause but water availability or subsoil

constraint. Finally, even in high-input systems, it has been conclusively demonstrated that the within-field varia of biomass production are related to topography-driven erosion processes (e.g. Reyniers et al 2006): this implies that erosion contributes to a decline in productivity, relative to non-eroding conditions, even when overall productivity increases. This is the focus of our paper and we therefore not consider changes in effort in our analysis.

The presented data do not result from an assumption about the relationship between soil erosion and biomass productivity but from observed cases in eroding landscapes under controlled amendment. We discussed these points section 2.1 (P3 L8, Data meta-analysis) and in section 4.1 (P17 L8, Model limitations).

I wonder why EPIC was not used. Doesn't this do essentially the same job but allows a better control of agronomic practices and all other parameters that influence yields (which all are completely ignored in the manuscript). EPIC would allow deriving yields from productivity. This also leads to the next influence that the authors do not consider: some causes of productivity decline by soil truncation are difficult to remove while this is easy for others. For instance the authors expect the largest effect on SOC decline from a loss of nutrients due to erosion (although this is pure speculation). Such a loss of nutrients would be easy and cheap to replace in many countries. Reversibility of productivity again points to the importance of the effort-yield relationship.

The main goal of the paper was to assess the potential impact of soil erosion on crop productivity and yield, assuming no changes in external inputs. Although we agree that it would be interesting to include the effects of agricultural management practices in the model, this is beyond the scope of our study. Furthermore, the required input data to constrain the spatio-temporal evolution of external inputs for the case studies (covering several decades) are simply not available. We could use EPIC but this would substantially increase the uncertainties associated with our model simulations. Finally, we agree that in intensively managed systems, fertilizer applications compensate for erosion-induced nutrient losses and that nutrient loss may not be the most important effect of erosion. Rooting space and water availability are more likely to be key issues. However, by representing different functional forms, we present all possible cases. We discuss this in the revised manuscript (P18L10 and P18L33).

In our study, we used the relationship between soil truncation (as a result of soil erosion) and relative yields published by Bakker et al. (2004) in a process-based SOC model. The simple model structure allows us to keep the number of input parameters in balance with the available data input. We also emphasize that the model used here has several advantages such as the detailed profile (i.e. with depth) - representation of SOC profiles, as well as its temporal evolution in response to erosion. However, we carefully revised the literature review on mechanistic SOC models, and further clarify the scope of our paper in the introduction (see P2 L16) and in the discussion (P17 L7, Section 4.1).

The basic relation between soil depth and yield is given in figure 1. This figure suggests that the study used data but this is misleading. In fact only one conceptual relation was used although the authors suggest that this relation can be separated into three different cases. My main critique regarding this figure is twofold:

(i) It ignores a fourth rather common case, namely that productivity first increases with increasing soil truncation (often up to a truncation of 20 cm to 40 cm) and then starts to decrease. This behaviour can be found in many loessial landscapes and the effect is so strong

that at least in former times without subsidies farmers paid higher prices for land where the clay depleted AE horizon had been lost and the better structured Bt horizon improved the properties of the Ap.

Figure 1 is based on data that were published in the review paper by Bakker et al. (2004) on "The crop productivity-erosion relationship: an analysis based on experimental work". This publication compiled data from 24 experimental studies, and they analyzed the effect of soil truncation on yield by comparing the yield to a reference yield. Following their review, we used a subset of these data that exemplify the relationship between soil truncation and yield based on field experiments from comparing paired-plots. In the dataset published by Bakker et al (2004), there is one case study (Olson et al 1999) where an increase in yield was observed as a result of soil truncation. This change was reported to be maximum 1.1 times the reference yield, and was observed after 25cm of soil truncation:

The Olsen et al (1999) study shows a decline in relative yield to 0.9 for the first 7.5 cm of soil truncation, followed by a slight increase to 1.1 relative yield. Furthermore, in the Belgian loess belt, Reynier et al (2006) studied the effect of soil truncation on yield and found that, even with soil amendment, yields were lower on the slopes than on the plateau as a result of soil truncation and removal of topsoil. Fenton et al. (2005), and Gregorich et al (1998) came to similar conclusions for sites in the US, and Dusar et al. (2011) and Kosmas et al. (2001) for the Mediterranean Region. The latter two studies indicated that, even with input of fertilizers, yields were decreasing, relative to stable parts of the landscape, as a result of soil erosion.

We agree with the reviewer that a yield increase is possible for specific cases but the available data suggests that it may not be representative for a more generally applicable soil erosion – productivity relationship. We discuss this issue in the revised manuscript on P17 L9 and P18 L10.

(ii) The interpretation of these three conceptual cases is brave. The authors explain a steep decrease in yield at little truncation by nutrient limitation. This is quite opposite to text book knowledge of plant nutrition. Since the early times of Mitscherlich we know from the law of diminishing returns that a reduction in nutrient availability has little effect when starting at high availability. For the topic of the manuscript it is completely irrelevant whether the one curve is caused by nutrient loss and the other curve is caused by loss of water holding capacity. These interpretations, which are repeatedly treated in the manuscript like truth although any proof is missing, should entirely be removed.

Based on the experimental data published in the meta-analysis by Bakker et al (2004), we have identified one mathematical expression that allows to express the change in relative yield as a result of soil truncation. The use of a simple mathematical expression facilitates its integration in the SOC dynamics model. Bakker et al. (2004) state that, following the literature, the three main regressors explaining yield losses due to soil truncation are water-availability, nutrient depletion and physical hindrance.

We agree with the reviewer (and the editor) that the interpretation of the three functional forms is not always straightforward. As this is not the main point of the paper, we revised the manuscript and removed these interpretations.

I would suggest that the authors strictly follow the rule of notation in mathematics. E.g.: sometimes they ignore the multiplication sign and AB means $A \times B$, in other cases AB means

one variable; sometimes variable are in italics, sometimes not; sometimes even mathematical signs are in italics (it should be dt). Units are similarly ambiguous (e.g., the unit coulomb is reported but not meant). I suggest following the "Guide for the Use of the International System of Units (SI)" (https://physics.nist.gov/cuu/pdf/sp811.pdf).

We apologize for this. We made the necessary corrections to the annotations.

The data that were used to calibrate the model come a bit out of the blue. "we used data from ten study sites" but I am not sure whether the five references distributed within this paragraph were the origin of the data. Without clear reference there is no information about their reliability and the boundary conditions under which they were carried out.

Our apologies for this confusion. In fact, the data that were used to calibrate the model were presented in Van Oost et al. (2007), and the characteristics of each site are summarized in the supplementary material of Van Oost et al (2007). To avoid redundancy, we referred to Van Oost et al. (2007). We modified the text and clarified the source of the source of the data in section 2.4 (P7 L24, citations correspond to the paper presenting each site, when available) and section 2.5 (P9 L3). Furthermore, we clarified how observed values of SOC losses and vertical fluxes were obtained from Van Oost et al. (2007) data in section 2.5 (P9 L3).

Some assumptions inherent in the model and some equations seem doubtful and would need better justification or modification:

(i) The model treats organic manure and plant residues identical (eqn 2). I wonder whether this is true because digestibility of fresh plant material is around 75%. Hence only 25% is left after the passage of the digestive tract and it is likely to assume that he remaining 25% are more resistant to further degradation than the initial material. Furthermore, in solid manure often stabilization processes take place that do not occur with plant residues on the field.

Manure and residues have a different humification coefficient values (respectively 0.3 and 0.125) which, in effect, leaves different amounts of C entering the first layer of the soil profile. These values result from parameter calibrations presented in the original ICBM model paper by Andren and Katterer (1997). We clarified the manuscript at P5 L16.

(ii) Surprisingly, the humification factor then distinguishes between manure and crop residues although this is not possible at this stage anymore because eqn2 has already mixed manure, crop residues and other young carbon into one young pool.

At each time step, the humification values are calculated based on the input from crop and manure at the considered time step, then the values of the C pools are computed. For the sake of clarity, the order in which the equations are presented in the text differs from the order of the calculations in the model. This probably caused the confusion and we clarified this in the revised manuscript at P5 L13.

(iii) The model considers only temperature as climate and edaphic (!?) factor (which temperature is not said), while usually soil moisture is the most dominant influence on SOC stabilisation (see Jenny 1941).

We argue that soil temperature is important for the C mineralization rate and the evolution of the SOC stocks. The ICBM model takes the moisture into account in the factor "r" (climatic factor) (Andren and Katterer, 1997).

(iv) The model does not consider any preferential loss of SOC or clay by erosion. The results may thus only be valid for tillage erosion.

We agree with the comment that we did not include selective erosion. However, based on several studies (e.g. Wang et al., 2010, 4 years monitoring), the observed enrichment is relatively small (1.3) which indicates that most of the erosion occurs under aggregated form, at least in fine-textured soils. We developed this issue in the discussion at P18 L10.

(v) Only roots incorporate SOC into subsoil. Bioturbation, leaching and other processes are omitted.

It is correct that our model does not take these processes into account. We argue that at a timescale of 60 to 200 years, SOC dynamics are largely dominated by soil redistribution processes and that bioturbation and leaching, although important processes on long timescales, account for a minor part of SOC fluxes and dynamics in this context (Doetterl et al., 2016, Minasny et al., 2015, Kirkels et al., 2014). We identified and discussed this shortcoming in a revised version of the manuscript in the point 4.1 "Model limitations" (P17 L7).

(vi) It is not clear, what follows in the model below 100 cm depth. I had the impression hard rock (i.e. the model does not shift the entire soil profile downward, when topsoil is lost). In this case, the model would be far too simple because hydrology then becomes tricky. Lateral water movement could not be ignored anymore when large parts of the soil were removed. Modelling would be easier and the results likely more realistic if soft rock would follow below.

We consider the following boundary condition: the soil properties (SOC, clay) observed at 1 m depth are representative for the soil/soft rock below 1 m. In the model implementation, SOC and clay content in the 100th layer are assumed to represent the soil characteristics below 1 m. Assuming a constant bulk density of the soil, soil characteristics are advected upward in response to soil erosion, proportionally to the amount of removed topsoil. As a response to soil erosion, the soil properties of the 100th layer are also continuously advected upward. In the physical hindrance case, the amount of coarse fragments is actually given by the low absolute clay content (in volume). We described this boundary condition more clearly in the revised manuscript at P7 L6).

(vii) Eqn (9) seems to be wrong because all carbon that leaves the young pool is delivered to the atmosphere although large part of this carbon (see humification factor) enters the old pool.

Equation 9 is simply the difference between the input of carbon entering the soil and the mineralized carbon leaving the young and old pools. Eq 9 is based on Fig. 1 from Andren and Katterer (1997) who developed the ICBM model. We checked Eq 9 with the original ICBM formulation (Andren and Katterer, 1997) and it is correctly presented in our manuscript.

(viii) A value of 0.55 or 0.6 seems to be more appropriate for alpha than 0.7. This could have considerable influence on the results.

The authors decided to use RRMSE for optimization (eqn (10)). Why? Isn't this a bad decision because it puts larger weight on layers with low SOC content although those layers are rather unimportant and relative measuring error is larger there? The authors also seem to have forgotten that they used RRMSE because they frequently report units of RRMSE (e.g. in Fig. 2) although this parameter cannot have a unit.

We argue that the shape of the SOC profile is as important as topsoil SOC content for our study. As Kirkels et al. (2014) pointed out, SOC stock and lateral fluxes follow a two-phase evolution in which the very high rate of loss in the first decades is followed by a period of lower loss rates. The evolution is similar for the vertical C fluxes as the C uptake increases fast at the beginning of the erosion period while the rate of increase is slowing down over time. This temporal evolution is due to the lower SOC content in the subsoil. Hence, the shape of the SOC profile determines the intensity and evolution of both lateral and vertical C exchanges. As these fluxes are a key part of our analysis, the RRMSE was used so that parametrization of the SOC profile would ensure a good representation of (i) observed SOC profile and (ii) an accurate representation of the impact of soil erosion on C fluxes. We added a short justification of the use of RRMSE at P8 L21.

Is a model error of 93% or even 121% acceptable (see Table 1b)? I would not be satisfied. Table 2a+b: How can the contribution of all parameters sum up to more than 100%?

The model error is indeed high for the cumulative vertical C fluxes, in contrast to the prediction of the SOC stock loss. We point out in the discussion that this discrepancy is mainly due to the fact that site-specific data is lacking to fully reconstruct the initial conditions and management options. Secondly, the long timescales considered should be considered when analyzing the model errors. Nevertheless, we would like to emphasize that the model predictions are in the correct order of magnitude and the relative differences between the sites are well represented.

In FAST analysis, the sum of contributions can be more than 1 when two (or more) variables are correlated. In our case, erosion rate and yield response to soil truncation are correlated. We clarified this point in the text at P13 L17.

Table 2b: How can erosion rate have an influence on the result although erosion rate was set constant?

A FAST analysis can show small positive contributions for constant parameters when (i) the number of runs is too small and (ii) due to mathematical dispersion. This also applies to negative contributions. We clarified this point at P13 L17 and in the table caption.

The Results chapter does not differ in style and content from the preceding chapters, which were assigned to Material and Methods. Most results are in fact reported in the preceding chapters. The manuscript requires better structuring

We substantially revised the manuscript and this has resulted in an improved separation of materials and methods from results.

Fig. 4: Units of the left panel? Shouldn't be a time unit in the right panel? What do the black lines denote?

The left panel represents the relative SOC loss $1 - \frac{SOC\ (final)}{SOC\ (initial)}$. It is thus without dimension.

Fig. 5+6 are in poor quality. Use the same font size as in fig. 4

This may be due to the compression applied to the file when submitting the manuscript. We provided figures with better quality.

Fig. 7: the information about the treatments is repeated three times (twice in the figure and once in the caption). What do the boxes and whiskers show (there is no convention on this)?

The boxes represent the interquartile range and whiskers represent the 95 % quantiles of the distribution. We corrected the information and added a description of the meaning of the boxes and whiskers in Figure 7 caption.

I didn't like the Discussion. What I missed at the very beginning is a paragraph about the assumptions and simplifications of the model and which influence they can have on the results (a little bit on this can be found at the very end but this is not stringent enough). Be more critical regarding your work. This would increase its value. At the moment it is of little value for me because I do not know under which conditions the results would apply and under which conditions nothing could be said. Studies are cited which seem to be in agreement with your results but this does not mean much. It only becomes meaningful if we know your assumptions and simplifications because then we also know that these assumptions and simplifications would not be important for the other study.

On the other hand there are parts in the discussion that could be written even without the preceding results (e.g. the last paragraph of chapter 4.1). They could be deleted in order not to increase the length of the discussion. Also all speculations about hindrance or nutrients should be deleted. They are all unsubstantiated and misleading.

We agree with this comment and we added a paragraph on model simplifications and assumptions in the revised manuscript (see section 4.1, P17L7). We also removed the less relevant parts of the discussion and modify the discussion as suggested by the reviewer.

Details:

In general, the use of blanks is strange. After semicolon the authors do not like blanks. Also periods are often omitted (e.g. in i.e.)

We corrected the typos and mistakes in the manuscript.

Figure 2 only allows for yield reduction. Yield increase would also be possible (as in the already mentioned case of alfisols or in the case where an acidified topsoil is lost; there may be more cases).

See discussion above. However, based on your comment, we discussed the implication of increasing yields on our results (P19 L4).

I wonder why the authors used different orientation of Table 1a and 1b. The same orientation in both parts would be possible. I suggest using the same orientation as in Table 1b also in Table

1a because this is the standard orientation (variables in columns, cases in rows). Table 1 b shows the vertical C balance. In all other cases this is called vertical C flux (at least I assume that this is the same). Be consistent.

We changed the orientation Table 1.

Fig. 3: Aren't the red profiles calibrated profiles (the word "simulated" would then be misleading). I thought the manuscript was about arable soils but apparently these soils do not have a plough horizon. Is the manuscript about grassland or woodland soils?

We agree that these are calibrated profiles and corrected the text. The study is only for arable lands, and we did not take tillage into account, only water erosion.

The manuscript frequently reports 1000 parameters. Fortunately the model has less. I guess the authors mean 1000 parameter sets.

We apologize for this, it is correct that we generated 1000 sets of parameters values. We carefully checked the manuscript and clarified it.

There are many more technical details (e.g. inconsistent tenses, omitted periods and blanks, inconsistent formatting of references) but given that large changes are necessary it does not make sense reporting these details.

We carefully revised the manuscript, and pay attention to the formatting of text, references and tables.

**2.2 Anonymous Referee #2**

GENERAL COMMENTS

The interdependencies of erosion and soil carbon balance have been investigated in many model-based studies. In a next step, vegetation should be explicitly included in integrated simulation approaches. Therefore, the authors tackle a relevant topic.

The general approach of the study is suitable to investigate the interdependency of erosion, plant growth and soil carbon balance. Nevertheless, the implications of the chosen implementation are not clearly addressed and are not sufficiently considered in the interpretation of the results. The study contains several flaws, which need to be addressed by the authors to make the results publishable in SOIL (see below).

In addition, the manuscript is not carefully prepared, hard to read, and hard to understand. This is mainly due to the lack of a common thread and to the fact that the authors use different terms for the same thing throughout the manuscript (e.g. net flux and cumulative flux and vertical flux, erosion and soil truncation, etc.). The mathematical notation is not clear and does not follow a general concept. Therefore, the text requires a complete revision to make it suitable for a scientific journal.

The necessary clarifications on terminology and mathematical notations have been made in the final manuscript.

SPECIFIC COMMENTS

In the following paragraphs, I address the main problems I found concerning methodology and presentation of the study:

The study applies a very simple SOC model together with an equation that relates yield to erosion and an approach to translate this into depth-dependent carbon input to the soil. From my point of view, this model must not be labeled "integrated", since that would require a plant growth model. Therefore, the title of the paper should be changed. The same applies to the statement the model would dynamically link crop yields, soil properties and SOC dynamics. The model does not contain a dynamic link from soil properties to crop yields but a static assumption on the effect of erosion.

The authors use two scenarios, one of which they call FB (feedback). However, Fig. 2 and the model description reveal, that actually there is no feedback loop in the model. Using the term "feedback" is therefore misleading. I suggest using a term like "yield effect".

A central point of the study is that agricultural yield changes with soil truncation. However, there is no direct link between these variables. Soil carbon input depends more on total biomass than on yields. However, the fraction of the harvested plant organs from the total biomass (harvest index) is physiologically controlled and therefore it is variable. As the authors point out, there can be different causes for the effect of erosion on plant growth. In the real world, farmers take measures to compensate for these effects. These simplifications need to be addressed when describing the general approach of the study and have to be included in the discussion. In this context, objective iii where the authors state their intention to investigate long-term effects of erosion on crop growth also needs to be rewritten.

In the current version of the model, there is no explicit link between soil properties and crop yields. Our study is based on a published relationship between soil erosion (expressed as soil truncation) and relative yield. The data about these relationships were obtained by field experiments comparing yields in eroding plots with yields in non-eroding areas. Hence, our data implicitly represents the aforementioned effects. The soil properties, SOC dynamics and C input (derived from yields/biomass productivity) are integrated as SOC dynamics depend on clay content and C input which are influenced by soil erosion. These links are therefore explicit. We thus reformulated carefully the title and the text so that the difference between explicit and implicit links are clear. The feedback term has been adapted to clarify that it represents the indirect effect of erosion on SOC dynamics through yield reduction.

We agree that yield and biomass production are two different concepts, which are often mixed in the literature and common language. In this paper, we are talking about biomass productivity in response to soil erosion and we agree that farming practices will try to cope with declining biomass production. We clarified it in the implementation (see i.e. P4 L1) and discussion that we are assuming constant agricultural management practices (P17 L7).

In the results, the authors present data on relative yield. Here, an explanation on how the reference value was set by Bakker et al. (2004) is missing. This is crucial in order to assess the results.

We selected data from comparative plots in which the original studies compared yield obtained in non-to slightly eroded soil with yield in eroded soil. Relative yields were calculated as following: relative yield is set to 1 for the non-or slightly eroded soil and fractions of that for yields on eroded soils. Hence, a relative yield of 1 indicates that there is no change in the yield, values < 1 represent yield losses and values > 1 yield gains. We clarified this in the text and figure captions (P3 L12 and P4 L3)

The model description is hard to understand because different terms are used for the same thing (e.g. input from crops vs. flux from the atmosphere). It requires a more precise presentation. In addition, the following points have to be addressed:
- the timestep of the model has to be given
- equation 2 and 3: If $h \neq 1$, where does $(h - 1)kyrY$ go? This is only implicitly stated in Eq. 9
- using 100 soil layers seems very detailed compared to the very general assumptions on vegetation effects and C input from roots. Why did the authors choose 1 cm for layer thickness? - Eq. 9: it should be stated, that this is just the sum of equations 2 and 3. One could factor out r, which would also simplify Eq. 3
- values for $\delta$, $kyt0$, and $kot0$ are missing
- Eq. 4 contains manure input, however there is no further information on this

The model time step is 1 year. We added this information at P4 L19.

Equation 2 and 3: the quantity $(1-h)*k*r*Y$ represents the mineralized/respired fraction leaving the young C pool. We added this information at P5 L9.

We used 100 soil layers to have a very fine representation of the vertical soil profile and advection in response to soil erosion. We found that the model was sensitive to the vertical SOC profile and using a coarse resolution resulted in substantial numerical dispersion and smoothing. In addition, as the model computational performance was very good, there is no need for a low vertical resolution. We added this information at P7 L2.

Eq. 9 is the sum of both equations. However, Eq 2 and Eq 3 are the classic way to present ICBM equations. We refer to Andren and Katterer, 1997 and SPEROS model presentations by Van Oost et al. (2005), Dlugoss et al. (2012), Nadeu et al. (2015).

Values for $k_y$ and $k_o$ are respectively 0.8 and 0.006 $yr^{-1}$ and $\delta$ is 2.91 (dimensionless). These values will be added to the manuscript. We added these values at P5 L9.

It also remains unclear, how soil truncation is modelled. Are layers removed from the top? Are the properties of the existing layers altered while keeping the overall soil depth constant? This has to be presented (considering the proposed effect of soil depth on plant growth).

Soil truncation is modelled by removing soil properties from the top of the profile. Assuming a constant bulk density, the considered depth does not change over time but the soil characteristics are advected upward in response to soil erosion. Soil properties are advected upwards in proportion to the amount of soil removed (see e.g. Van Oost et al, 2005, Dlugoss et al., 2012, Nadeu et al. 2015). At the bottom of the profile, a constant boundary condition is assumed and its properties are progressively included in the soil profile proportionally to the amount of removed topsoil, resulting in an effective truncation of the soil profile characteristics. We clarified how the vertical transfer is represented at P7 L1.

The next point concerns the model validation. As far as I understood the text, the same observational data was used for validation and calibration. If I got this wrong, clarifying text has to be added. If I am right, this is not a validation but an evaluation. Nevertheless, a validation is required and can be accomplished by using a leave-one-out or bootstrapping approach. As I also commented in the context of the long-term experiment, the validation needs to be conducted with the same set of perturbed parameters as the following experiments. The text states that Fig 3 shows a comparison of simulated SOC content and observations. This comparison should also include uncertainty information resulting from the 1000 simulation runs with perturbed parameters.

We share the concerns raised by the reviewer: we realize that we have not sufficiently explained how the model calibration and evaluation was implemented. We would like to emphasize that we do not calibrate the model parameters on observed SOC losses or soil-atmosphere SOC exchange. We simply fitted the three model parameters that control the shape of the SOC depth profile on stable sites only. This procedure therefore only estimates the initial conditions of the model for each site and should not be considered when evaluating the performance of the model. This is also the reason why we did not include uncertainty ranges in figure 3 as only a single profile was available for each site. In a second phase we evaluate the model using observational data on SOC losses and soil-atmosphere exchange in response to erosion. Importantly, we did not use this data to inform/calibrate the model. We therefore believe that this represent a robust way to evaluate/validate our model. We have adjusted the text to make this approach clearer. We performed site-specific simulations as SOC parametrization, clay content, erosion rate and length of the simulations were specific for each site (see Van Oost et al. (2007) paper): however, these are estimates and are associated with substantial uncertainty. To address this issue, we performed an uncertainty analysis: For each of the 10 sites, we created a set of 1000 scenarios for which parameter values were randomly chosen in a narrow range around their published values in Van Oost et al. (2007). These values, associated ranges and lengths of simulations are given in Table 1a. We clarified the calibration/parametrization procedure in P7L25 and better separated the calibration description from the evaluation description (P9L2) to avoid confusion.

After the model validation, the reader will be interested in results of the model runs. How does SOC and C-exchange with the atmosphere develop over time? The authors should present timeseries that enable the reader to get an idea of how the model works. If the data were available, a comparison to observations would be desirable.

The time-series resulting from our simulations are available and could easily be included in the paper. SOC stock evolution follows a classic two phase evolution: the profiles quickly lose a large amount of carbon during the first decades, and the rate of SOC loss is then decreasing over time due to the lower SOC content of the exposed subsoil. When the yield effect is weak, a steady-state is observed whereby the laterally exported SOC is replaced by new C coming from plant inputs. When the yield effect is strong, it takes a longer time to come to steady-state SOC stocks or there is no steady-state. However, due to the large range of simulations performed (1000) with a large range of parameters, it is rather difficult to visually synthetize the information into a graph (see figures inserted below). Furthermore, to our knowledge, long-term observational data on yearly C fluxes nor SOC stock evolution are not available in literature. Hence, we chose not include it for clarity.

[Figure]

SOC stock evolution (t/ha) for the FB dataset. Solid line denote the mean of the 1000 simulations, shaded areas represente one standard deviation (dark grey), two standard deviations (middle grey).

[Figure]

Annual vertical C fluxes (kg./m²) for the CTL datase (blue) and for the FB dataset (grey). Postive value represents a net C capture from the atmosphere to the soil, negative values represents a net C emission from the soil to the atmosphere. Solid lines denote the mean of the 1000 simulations, sahded areas represente one standard deviation.

Concerning the long-term experiment, it remains unclear, why a second set of perturbed parameters was generated. In order to evaluate the results, the experiments have to be conducted with the validated model and the same sets of parameter values. In addition, information on the scientific basis of the choice of value ranges for the parameters is missing. This is of great importance if the intention of the FAST analysis is to compare the tested parameters regarding their influence on the overall variability. This is because the value ranges used for the parameters have an effect on the resulting explained total variance. In order to interpret the FAST results in the way the authors do, it has to be argued why the value ranges are comparable. Using the same relative ranges is not appropriate due to different relative ranges of the respective parameters in the field. An appropriate method is to use published ranges of observed values together with estimates of uncertainty. If these are not available, reasons for the estimates of plausible ranges have to be given.

The model validation was done comparing the model predictions against observations using site-specific data. These data are displayed in Table 1 and Figure 3. We added a relative uncertainty range around these observations to account for natural variability and errors in measurements at the site-scale. A range of B exponent was attributed to each site, in line with each site's description of soil depth description and climate type. For each individual site, we generated 1000 sets of parameters, which values were inside the range of this specific site. We performed 1000 simulations which time length was site-specific (i.e. 1000 simulations with the parameters of Belgium 1 site, 1000 simulations with the parameters of UK site, etc.). Therefore, the resulting SOC losses and vertical C fluxes can directly be compared to the observed values as the erosion and SOC parameters were close to the observations.

The long-term experiments should be considered as an exploration of the model behavior at longer time-scales. We therefore performed a numerical long-term experiment on the total range of the observed parameter values regardless of the sites considered in the model evaluation: i.e. from the smallest value to the highest value found in the table, with the notable exception of erosion rate, which range was extended further based on erosion data across Europe and the USA. We generated 1000 sets of parameters based on this total range of values (as presented in table 2). Specifically, the range of the yield-effect exponent was chosen to cover the whole set of yield values per unit of soil truncation as extracted from Bakker et al. (2004) and this was presented in the first part of the manuscript. The root-depth parameter indicates the root penetration in the soil and its value was taken so that 95% of the roots are distributed in the first 35 cm to 65 cm with respective φ values of 4 to 6, with 30 to 45% in the first 20 cm. These values are in accordance with previous SPEROS parametrization obtained by inverse modelling (Dlugoss et al., 2012, Nadeu et al, 2015). As for the mineralization distribution, the given range indicates a turnover rate at 1 m depth of 137 to 700 years for the slow C pool which is in line with the centennial turnover rate found in deep colluvium by Wang et al. (2014) or Van Oost et al (2012).

We thus argue that the interpretation of the SOBOL/FAST analysis is valid and we will more clearly identify in the text where the ranges of the parameters come from.

We better explained how the dataset were built and in which simulations they were used in section 2.5 (model evaluation, site-specific datasets) and 2.6 (long-term experiments, extended-range datasets).

It also is not clear to me, which set of model runs was used for the analysis in sections 3.3 and 3.4. Is this based on the same results as the FAST analysis?

The results for the long-term simulations (200 years) in section 3.3, 3.4 are based on a set of 1000 scenarios randomly chosen in the range of values specified in Table 2a. This set was also used in the FAST analysis. We better clarify the use of each dataset in sections 2.5 and 2.6.

Finally, the study requires a comprehensive discussion on the transferability of the results to the real world. Especially the implications of the simplifications in the model on the transferability have to be dealt with. In addition, the authors should discuss the role of the farmers adjusting their choice of crops, management practices and harvest residuals, etc. This is tackled shorty in the final sentences of the discussion, but this is not sufficient. Other important points to be discussed are the dependency of yield on plant growth, on nutrient availability, and on access to water. All this can alter the harvest index and therefore the relation of soil carbon input and yield.

We added and clarified the aspect related to the model limitations and the agricultural practices adaptation in the extended discussion about the study limitations (see section 4.1 and P17 L23). As for the dependency between yields, nutrient availability or water availability, these agronomic aspects have been discussed abundantly in the literature (e.g. Bakker et al in their review (2004), Christinsen and McElya (1988), Lal et al. (1999) or Larson et al. (1985)). However, following the comments of Reviewer 1 and the editor, we removed the interpretations of yield reactions to nutrient limitations, soil depth or water availability to focus on the mathematical form, except in section 4.2 (discussion). Furthermore, we consider that a more detailed analysis of the biological effects of soil truncation of plant growth is outside the scope of this paper.

In the beginning of the discussion, results are compared to Berhe at al. (2005), which, in contrast to the present study, found a carbon sink. Explanation is required why this is rated as a support of the new results.

Berhe et al (2005) found a carbon sink related to the C uptake from the atmosphere occurring in eroding areas. Our study found that erosion can result in a carbon sink (in terms of vertical C fluxes) as the balance is often positive with C being added to the soil. Our study however emphasizes that this C uptake can be overestimated in modelling studies if the long-term evolution of the yields is omitted.

In the final paragraph, the authors reveal, that with B>1.1 there was no effect on yield. If this is the case throughout the study, the manuscript can be simplified by stating this in the beginning and removing this aspect in the results section.

We refer to our reply to reviewer#1. The main goal of our study was to explore the effect of biomass productivity decrease on SOC losses and vertical C fluxes. We acknowledge that the model is relatively simple and required assumptions about the relationship between the C input and the biomass productivity. We agree that the relationship between C input and biomass can be dependent on the amount of residues left on the field but under the absence of data, it is difficult to correctly represent this. When B is larger than 1.1, the simulation period was not sufficiently long to push the system with a heavily convex relationship to the tipping point, leading to a relatively low response of the C stocks and vertical fluxes to the addition of such a feedback.

DETAILED COMMENTS AND TECHNICAL CORRECTIONS

We thank the reviewer for the suggestions. For the sake of clarity, we only answer individual comments relative to understanding, clarifications, and precisions. All other comments about typo, references or re-phrasing which do not required detailed answers will be addressed in the revised manuscript.

Use the same font and italics for symbols in equations and text unless there is an explicit rule given by the journal.

The journal asks for equation symbols to be in italic when used in the text. The necessary changes were made.

Improve the graphical quality of the figures.

We improved the quality of the figures.

p1l22: why negative numbers for an increase in SOC losses?

This is a mistake as the numbers represent the relative C stock changes. We corrected it.

p3l14: this is a meta-analysis, not an analysis of meta-data

We agree with this comment changed it throughout the manuscript.

p4l3-4: unclear why a clay-fraction can replace explicit accounting for soil depth

The clay fraction can be given in absolute terms, i.e. the volume of clay in a given volume of substrate (soil + rock fragment) or in relative terms as the fraction of clay in the remaining space, not occupied by rock fragments. In our case, the clay fraction is accounted for in the model by the absolute volume of clay per volume of substrate. We further assumed that the relative fraction of clay in the remaining space and the bulk density of the soil are constant. Hence, the absolute amount of clay indirectly indicates how much rock fragment is contained in the substrate, which is a proxy of soil depth as 100% of rock fragment is representing the bedrock level. In the case of deep soft rock, the absolute clay content shows little variation between the topsoil and the bottom of the profile. In the case of physical hindrance, the clay content is highly reduced at the bottom of the profile. We provided a better explanation of this simplification in section 2.1.

p5l1: there are two van Oost 2005 papers in the references. Please specify. The same applies to some references to van Oost et al. (2007)

The references were corrected.

P5l26, Eq. 7: K has to be lower case since rates were introduced lower case in Eqs. 2 and 3. In Eqs. 4, 6, and 9 dependency on time and/or layer is denoted by t and z in parentheses. Therefore: k(t, z). In addition: explain to the reader that this is used for ky and ko p5l27: Sentence incorrect
p5l30: refer to equation 4

Changes have been made, and ky and ko were defined (P5 L9)

p5l31: to make it easier for the reader to understand the overall model setup, state the source of the cumulative soil truncation data

We added the information about the link between erosion and cumulative soil truncation at P3 L27 (which is the annual erosion rate * time, as erosion rate does not vary).

p7l9: two instead of 2
p7l13: remove second full stop

We corrected this.

Table 1 a: What does "period of cultivation"mean? A single number does not define a period.

The period of cultivation is the total duration of cultivation between the start of cultivation on the considered field and the date of the final analysis. We corrected the header to "time since start of cultivation" in table 1.

Table 1 a: the caption mentions data for two simulated scenarios, which cannot be identified in the table. In addition, site description and results should not be in the same table

The caption was wrong and has been corrected accordingly to the content of the table.

p10l2: where do the years come from? Were these the same for each site? Is this somehow connected to the periods in table 1?

In this case, these are different from the period of cultivation as the $^{137}$Cs was released in the atmosphere and deposited after the nuclear bomb testing. In the literature and following Van Oost et al. (2007), we took 1954 as the standard date of $^{137}$Cs deposition on the earth surface (Ritchie and McHenry, 1990). This date is considered to be identical for all sites. As the erosion rate derived from $^{137}$Cs tracer were valid for the period post-1954, the integration of cumulative vertical C fluxes was done over the period from 1954 to the date of the C inventories realized in each individual site rather than over the entire period of cultivation. We added these details in section 2.5 (P9 L4)

p10l3: what is the "period of interest"?

This is the period of cultivation for SOC losses or the period between 1954-date of sampling for the vertical fluxes. We clarified the manuscript (P9 L6)

P10l5: the parameter sets were not obtained by calibration. This only applies to the mean values.

Correct, we changed the text.

P10l10: You investigate the feedback effect in the model. This is not a potential effect. Only transferring it to the real world makes it potential.

Correct, we adapted it.

P10l10: what does the "c." mean?

c. stands for "calibrated" years.

Table 2: use the same symbols as in the text; φ was introduced as a carbon input profile, not a root density profile. This also applies to p13l13 and p13l16

We will check the use of the symbols.

p11l11: instead of "typical values", state how the numbers were computed

These numbers of SOC losses were obtained by calculations based on the data provided by Van Oost et al. (2007): stable profile SOC stock, lateral SOC fluxes, vertical SOC fluxes and erosion rate for each site. The total C losses was calculated by integrating lateral SOC fluxes, vertical SOC fluxes and calculating a mass balance to obtain the total SOC lost over the cultivation period. The observed relative SOC loss is the ratio between the total SOC loss and the observed SOC stock. We clarify the method in the manuscript.

Figure 4: consequently use upper or lower case letters to address the graphs of the figure

We will adapt this.

p12l1: the highest observed SOC loss is said to be 0.19. However, in Fig 4 a, the highest red circle is slightly above 0.2.

We apologize, this is a mistake in the text.

Figure 5: If the same variable is on both y-axes, the axis labels have to be the same. Figures 5 and 6: When comparing simulation and observation or results from different scenarios, the graph should be square.

We made the necessary changes.

P19l5: Bouchoms et al. (2017) missing in list of references -> use a reference managing software to avoid this

We carefully checked the formatting of references, and make sure that the list is complete.

P19l6: this is a nice explanation of the possible interaction of processes. The authors should consider presenting this in the introduction.

Thanks for this suggestion, however, we think this explanation does not integrate well in the introduction logical development.

P19l20f: sentence unclear

We clarified the sentence which is describing the three processes involved into the C sink resulting from erosion (P21 L15).

[revised manuscript text omitted]

---

## Editor Decision (ED1)

The manuscript was substantially reworked and improved. Overall, I think it is a very interesting paper taking the decline in crop productivity following erosion into account, when simulating erosion-induced changes in SOC and vertical C fluxes. This is the logical next step to improve coupled C balance and erosion models. Nevertheless, some substantial improvements are still necessary before being accepted for publication.

General comments:

1. As Rev#1 pointed out that the authors should make clearer how they derive C inputs by plants from data of crop productivity or yield. Moreover, the terms should be use more clear. The terms crop yield, crop productivity and crop biomass are still not used clear enough. Firstly, I suggest to go through the entire paper and proof that either the term crop productivity (including a definition what exactly this means) or crop yield is used. Avoid any other terms. Secondly, more details are needed how to derive C inputs from crop yields (are all C inputs varied according to erosion-induced variability of yields (plant residues, roots))? Is the root-length-density affected by smaller yields (not modelled but needs to be discussed) etc..

2. Figure 1 and Equation 1: I have some general doubts regarding the use of the Bakker et al. (2004) data (Figure 1) to derive Equation 1. (1) The authors state they took alpha as -0.7 from Bakker et al. (2004), but this does not fit to the data presented in Figure 1. If the linear version of Eq. 1 (B=1) is fitted to the data (while forcing the y-value to 1 in case of no soil truncation) an optimal fit would result in an Alpha of about -0.4. With an alpha of -0.7 the data presented are only weakly described. So, it is not clear why this value was used. (2) Rev#1 rightly pointed out that the data presented in Fig. 1 do not allow to classify the different forms of Eq. 1 to represent water limitation, nutrient limitation and physical hindrance. So, the authors removed this classification but still used the same different equation parameters (varying B to get concave, linear, convex relationships). I do not see the added value of the different relationships not clearly supported by the presented data. To be clear here: The variation of B is not statistically supported by the data in Fig. 1. This would be only the case if you could produce sub-sets of the presented data (based on objective parameters). The presented variation of B to include more or less all data in Fig. 1, while using a low and a high value of B, seemed to be arbitrary. **I strongly suggest to use only one (linear) relationship which is data driven (you could still do a uncertainty analysis slightly varying the parameters of the equation). This would make the interpretation of the results less speculative**.

[Figure]

Fig. 1. Illustration regarding the sensitivity of your different forms of the Eq. 1.; Data taken form the manuscript; Black lines represent Eq. 1 as implemented in the paper. Blue lines based on a fit of the linear form of the Eq. 1 [(1+(-0.42*x^1)]. This illustrates that only the linear equation in blue is data based while all other forms are not supported by the data given in Fig. 1.

3. Rev#2 stressed that the parameterisation, calibration and evaluation (validation) are not clearly described. The manuscript has improved but still lacks clarity, especially regarding the comparison with 'observed' cumulative C flux data from Van Oost et al. (2007).

In chapter 2.3 the authors state that data from Van Oost et al. (2007) were used for parameterisation and validation. Following this first statement data from different references (which are summarized in Van Oost et al. (2007)?) are given. However, later in the chapter the authors mentioned "The form of the erosion rate-production relationship for each site was derived from the information presented in the original experimental studies". Which original studies do you mean here? Yields measured were not presented in Van Oost et al. (2007)? This is confusing.

In chapter 2.4 the authors state: "We performed a model evaluation using empirical observations on SOC losses and cumulative vertical C fluxes (Table 1) (Van Oost et al., 2007)." This is again confusing or misleading: (1) It should explained in more details that not the same data used for model calibration were used for model evaluation. (2) What are empirical observations on cumulative vertical C fluxes? For me it implies fluxes were measured, which was not the case in Van Oost et al. (2007). Erosion-induced vertical C fluxes were calculated based on measured erosion (137-Cs) and measured changes in SOC stocks. The same unprecise description can be found at the beginning of chapter 3.1. "In a second step, we evaluated the model based on the results obtained after the site-specific simulations by comparing the SOC losses and cumulative vertical fluxes to the **observed losses and fluxes** derived from Van Oost et al. (2007) data." I suggest either being more precise, e.g. "observed SOC losses and calculated erosion-induces vertical C fluxes" or just evaluate the results of the modelling against the observed data of Van Oost et al. (2007).

Chapter 3.1. Model evaluation (Fig. 4 b): This again shows how problematic it is to treat the Van Oost et al. (2007) results regarding cumulative vertical C fluxes as observations. The fluxes in Van Oost et al. (2007) were not observed but calculated from changes in SOC stocks and long-term mean erosion. So, you are comparing your model results with other "model results" which makes it difficult to judge if your simulation of vertical C fluxes improved or not. Again, I strongly suggest to critical rethink the comparison with the cumulative fluxes of Van Oost et al. (2007) or at least be more explicit about the data you use as reference for your model results. You might improve your paper if you do not compare with cumulative fluxes from Van Oost et al. (2007) as this leads to the questionable reduction in your model quality as suggested from Fig. 4b.

You find some more specific comments to your manuscripts on the next pages.

[revised manuscript text omitted]

---

## Author Response (AR2)

**Reply to the Editor**

The manuscript was substantially reworked and improved. Overall, I think it is a very interesting paper taking the decline in crop productivity following erosion into account, when simulating erosion-induced changes in SOC and vertical C fluxes. This is the logical next step to improve coupled C balance and erosion models. Nevertheless,
5   some substantial improvements are still necessary before being accepted for publication.

We thank the reviewer for these positive feedbacks and we would like to thank you explicitly for the many helpful and very detailed suggestions. They have improved the manuscript substantially. Additionally, you will find the authors reply to the individual comments in blue.

General comments:

10   1.   As Rev#1 pointed out that the authors should make clearer how they derive C inputs by plants from data of crop productivity or yield. Moreover, the terms should be use more clear. The terms crop yield, crop productivity and crop biomass are still not used clear enough. Firstly, I suggest to go through the entire paper and proof that either the term crop productivity (including a definition what exactly this means) or crop yield is used. Avoid any other terms. Secondly, more details are needed how to derive C inputs from
15   crop yields (are all C inputs varied according to erosion-induced variability of yields (plant residues, roots))? Is the root-length-density affected by smaller yields (not modelled but needs to be discussed) etc…

We have revised to manuscript accordingly, we now clearly define the terminology and use it consistently throughout the manuscript. The estimation of C inputs from crop yields are described (P67L1). It is correct that the model considers an equal impact of erosion on roots and plant residues. Due to the lack of observational data
20   to constrain a response model, the root-depth-density relations are assumed to be constant in our study. This is reported and discussed at P6L17 and P19L15.

2.   Figure 1 and Equation 1: I have some general doubts regarding the use of the Bakker et al. (2004) data (Figure 1) to derive Equation 1. (1) The authors state they took alpha as -0.7 from Bakker et al. (2004), but this does not fit to the data presented in Figure 1. If the linear version of Eq. 1 (B=1) is fitted to the data
25   (while forcing the y-value to 1 in case of no soil truncation) an optimal fit would result in an Alpha of about -0.4. With an alpha of -0.7 the data presented are only weakly described. So, it is not clear why this value was used. (2) Rev#1 rightly pointed out that the data presented in Fig. 1 do not allow to classify the different forms of Eq. 1 to represent water limitation, nutrient limitation and physical hindrance. So, the authors removed this classification but still used the same different equation parameters (varying B to get concave,
30   linear, convex relationships). I do not see the added value of the different relationships not clearly supported by the presented data. To be clear here: The variation of B is not statistically supported by the data in Fig. 1. This would be only the case if you could produce sub-sets of the presented data (based on objective parameters). The presented variation of B to include more or less all data in Fig. 1, while using a low and a high value of B, seemed to be arbitrary. **I strongly suggest to use only one (linear) relationship which**
35   **is data driven (you could still do a uncertainty analysis slightly varying the parameters of the equation). This would make the interpretation of the results less speculative**.

Thank you for these valid comments and suggestions. As indicated in the resubmission, we agree that the interpretation of the response form is difficult as many factors interact and the manuscript was modified

accordingly. The different response curves were discussed at length in Bakker et al (2004) and although there is substantial scatter, there is clearly a conceptual basis to use three different response curves (linear, convex and concave). This is discussed in very detail in the Bakker et al publication. Although we agree that the number of studies is small, we therefore argue that using on single regression through the scatter is not very useful in the context of our study.

According to the editor and reviewer suggestions, we modified the equation with a new leave-one-out cross validation and adapted the α parameter to 0.42. Simulations were re-run with the new parameter.

We would like to emphasize that our study does not aim to simulate the very specific local conditions of the experimental data represented in Figure 1. Our study is a numerical experiment to assess the effect on C storage over longer timescales based on several realistic scenarios. Although we accept that the interpretation of the factors controlling the different response curves is problematic, we do think that the use of different scenarios is very informative and provides a quantitative basis to assess possible effects of erosion on longer-term C stock evolution. We do not report a single scenario, representing the best fit through the data, as being more likely, and the model results therefore represent the possible range of SOC trajectories under the impact of erosion. In order to assess this comment, we modified and clarified the approach in P3L24, P8L20.

3. **Rev#2** stressed that the parameterization, calibration and evaluation (validation) are not clearly described. The manuscript has improved but still lacks clarity, especially regarding the comparison with 'observed' cumulative C flux data from Van Oost et al. (2007).

In response to this comment, we have further detailed the description of the model evaluation (see description below).

4. In chapter 2.3 the authors state that data from Van Oost et al. (2007) were used for parameterization and validation. Following this first statement data from different references (which are summarized in Van Oost et al. (2007)?) are given. However, later in the chapter the authors mentioned "The form of the erosion rate-production relationship for each site was derived from the information presented in the original experimental studies". Which original studies do you mean here? Yields measured were not presented in Van Oost et al. (2007)? This is confusing.

The 2007 paper only reports on the SOC fluxes, and although the effect of C input decline as a result of erosion is included in the estimations (due to the space for time substitution method), this was not discussed/analyzed. Since we use a very different approach where each process is represented in a time-explicit model, we need to identify the erosion-productivity relation explicitly. We provide a realistic approximation of the relationship for each site. This is now clearly described on P8L28 and P9L4. We hope that this addition improved the readability.

5. In chapter 2.4 the authors state: "We performed a model evaluation using empirical observations on SOC losses and cumulative vertical C fluxes (Table 1) (Van Oost et al., 2007)." This is again confusing or misleading: (1) It should explained in more details that not the same data used for model calibration were used for model evaluation. (2) What are empirical observations on cumulative vertical C fluxes? For me it implies fluxes were measured, which was not the case in Van Oost et al. (2007). Erosion-induced vertical C fluxes were calculated based on measured erosion (137-Cs) and measured changes in SOC stocks. The same unprecise description can be found at the beginning of chapter 3.1. "In a second step, we evaluated the model based on the results obtained after the site-specific simulations by comparing the SOC losses

and cumulative vertical fluxes to the **observed losses and fluxes** derived from Van Oost et al. (2007) data."
I suggest either being more precise, e.g. "observed SOC losses and calculated erosion-induces vertical C
fluxes" or just evaluate the results of the modelling against the observed data of Van Oost et al. (2007).

Only the parameters describing the initial SOC profiles required calibration; no data was used to calibrate the SOC
losses and vertical fluxes. Secondly, we would like to point out that the estimates presented by Van Oost 2007 are
indirect measurements of vertical C fluxes (differences in SOC stock over time) and did not involve the use of a
process model nor a representation of soil atmosphere exchange.

However, in order to address the issue and potential confusion raised here, we have removed the adjectives
'empirical' and 'observational' to describe this data. We hope that this makes it more digestible. Also note that the
approach to derive these estimates is clearly described in the methods so the reader has a better understanding of
how they were obtained.

6. Chapter 3.1. Model evaluation (Fig. 4 b): This again shows how problematic it is to treat the Van Oost et
al. (2007) results regarding cumulative vertical C fluxes as observations. The fluxes in Van Oost et al.
(2007) were not observed but calculated from changes in SOC stocks and long-term mean erosion. So, you
are comparing your model results with other "model results" which makes it difficult to judge if your
simulation of vertical C fluxes improved or not. Again, I strongly suggest to critical rethink the comparison
with the cumulative fluxes of Van Oost et al. (2007) or at least be more explicit about the data you use as
reference for your model results. You might improve your paper if you do not compare with cumulative
fluxes from Van Oost et al. (2007) as this leads to the questionable reduction in your model quality as
suggested from Fig. 4b.

We do not fully agree with the assertion that the 2007 data has little scientific value as a reference. Again, the data
is not a typical model-based estimate, but is an indirect measure of vertical C exchange (see the carbon inheritance
index). The fact that there is uncertainty associated with the reported values and that a simple mass-balance equation
was required does not make this a 'model'-based estimate. However, as indicated above, we have removed the
'observational' label to avoid any confusion. The approach proposed in the 2007 paper can therefore not be
considered as a model prediction, as it is firmly grounded in data (both SOC and erosion) and is in essence nothing
more than a space-for-time substitution. As a result, we do think that comparing model predictions, as we provide
here in this paper, with these indirect estimates derived directly from observational data, may provide a basis for
evaluation. Finally, we agree that the model performance is not good, but the main trends as well as the order of
magnitude of the fluxes were well predicted (as shown in figure 4b). Accordingly to your suggestion, we removed
the comparison of the cumulative vertical C fluxes and limited it to the SOC losses.

**Detailed comments**

We thank the editor the helpful detailed comments. Here is our reply to the specific comments made in the text.
However, for comments about the crop productivity-erosion equation please refer to the reply to the general
comments.

P2L2-L9 Here at the beginning of the paper it would be a good place to define what you mean with crop yield, crop
productivity, biomass production etc.

We simplified the terminology (yield, crop production, productivity, etc) and defined the meaning of the terminology "crop productivity" at P2L4.

P2L13 Redistribution is no loss per se. Typically lateral losses are understood as losses to the stream network.

We rephrased and made clear that it involves transfer to the fluvial system.

P2L16 What do you mean by transport?
(i) Transport during erosion event (scale of small watersheds?) or (ii) transport from arable land into streams and lastly to the ocean?
(i) There is nearly no loss during the transport within a small catchment (which last for several hours).
(ii) The transport to the oceans result in mineralization as show by several studies (e.g. Battin at al....)

We meant transport from erosion sites to deposition sites. The sentence has been modify accordingly.
We acknowledge that the effect of a weakened physical protection is a process occurring during transport is small but in the context of the paragraph describing the general processes of C exchange during erosion phases, we feel it should be mentioned.

P2L30 if you refer to a single approach 'CENTURY' you cannot state in the next sentence that the 'aforementioned models.

We clarified the sentences.

P3L4 Unclear! Are you referring to modelling studies in general or to experimental studies focussing on the decline of biomass productivity (such studies are available).

We clarified the sentence as we refer to the modeling approach and the long-term effects of crop productivity decline on SOC losses and vertical C fluxes.

P3L5: As rev#2 points out your model is not really dynamically linked. So I would avoid this here.

We rephrased the sentence and removed the term "dynamically".

P3L15: Bakker et al. (2004) stated: "Deterioration of soil by erosion often results in decreasing crop productivity. The extent to which crop yields respond to soil erosion depends on several variables such as crop type, soil properties, management practices and climate characteristics." Hence, they used the term crop productivity synonymous to crop yield. So, replacing crop yield (as indicated in the first version of the paper) with crop productivity as understood by Bakker et al. (2004) does not make any difference. As mentioned above be more precise with the used terms to avoid confusion.

As mentioned above, the terminology has been clarified and defined at P2L4.

P3L25 and P4L5 This reduces the number of studies to 6 (if I counted correctly in Bakker et al. (2004)), an information which needs to be provided, to show that the data basis is not very strong.

We agree that empirical basis is small and that a general response is not applicable. However, as stated in our reply to the general comment, on the one hand, these data provide a range of possible trajectories and (ii) our study is, in essence, about numerically exploring this range and its impact on C losses and vertical C fluxes. We clarified these points at P3L25.

P6L1 You ignored the comment of rev#1 that moisture might be more important than temperature. I suggest giving a bit more detail here (see original paper of Andren and Kätterer)

The requested precisions about the role of moisture in the r factor were added.

P6L19 Here you could easily add that aspects of bioturbation etc. are not taken into account (see comment of rev#1)

We clarified and added the precision about processes involved or not in vertical C distribution and transfer in the soil profile (bioturbation and leaching).

P7L1 This makes not clear how C input is estimated? Just multiplying all the estimated C inputs (via roots, residues, manure etc.) with the relative yield at a certain raster cell?
Be more explicit here.

The effect of the crop productivity decline on C input is given by Eq. 8, in which, C input become time-depent as they are multiplied by the relative yield obtained by Eq.1 at a time t. We clarified that only C input from crops were affected (residues and roots).

P7L10: What is new about the humification coefficient?

As the humification coefficient depends on C inputs (Eq. 4), it is now time-dependent and updated each year in response to changing C inputs in response to soil erosion and crop productivity decline (Eq. 8). We clarified this at P5L29 and P7L14.

P7L17: The model has a lot of smoothing effects due to its parsimonious structure. So, this is not a clear argument to use 1 cm layers.

This very fine representation of the vertical soil profile and advection in response to soil erosion is required due to sensitivity of the model to the vertical discretization as a coarse resolution typically results in substantial numerical dispersion and smoothing of the evolution of C fluxes between layers over time. We added these precision in the manuscript at P7L17.

P9L6: This is still the argument from the original paper that different types of erosion-productivity equations describe different processes. See general comments.
Moreover, from this it is not clear if the data used are from Van Oost et al. 2007 as all the other references mentioned here or from Bakker et al. 2004.

Based on information reported in the original studies, we tried to estimate the local erosion-productivity relationship for each site. Clearly, the erosion-productivity relations are generally poorly constrained and we therefore use a range for parameter $B$ to represent uncertainty. We modified the terminology (nutrient, etc) to the one used through

this version of the manuscript and made the necessary clarification about the choice of erosion-productivity for each sites in P9L4.

P10L1 As one of the revs. point out, it is not clear if you use the same data as used for calibration here for evaluation. I guess not, but this must be more clearly elaborated.

We did not use the same data for calibration and evaluation.

We use the environmental parameters (temperature, clay content, non-eroded SOC profile) provided by Van Oost et al. (2007) for each site to calibrated the C inputs, root-depth distribution parameter and the depth attenuation of C mineralization to obtain the initial SOC profile used by the model, before the simulations were run. This is explained at P8L15 in the calibration section. We took into account the uncertainty in the environmental parameters provided by Van Oost et al. (2007) and allowed them to vary in a range around the reported value.

Model performance evaluation was based on the results after the simulations by comparing modeled SOC losses for each sites with the SOC losses obtained by Van Oost et al. (2007) using observed profiles and space-for-time substitution. We assessed and compared the performance of the model with the erosion-crop productivity link (FB) and without the link (CTL). This is explained in the section "Model Evaluation" at P10L4.

We clarified these remarks in the manuscript.

P10L4 [SOC losses and C fluxes reported by Van Oost et al. 2007] Be more explicit here. What do you mean by reported. See also general comments.

Please refer to the reply to the general comment 4 and 5 about the data of Van Oost et al. 2007. We clarified this at P10L5.

P11L8 [Table 3]. This extended range is dictated by the difference between the smallest and largest values provided by Van Oost et al. (2007)] unclear sentence.

We took the maximum and minimum value of each parameter reported in the site characteristic compilation of Van Oost et al. (2007) to set the limits of the extended ranges. We modified the sentenced accordingly.

P12L5 The pools used in ICBM are virtual pools which cannot be measured directly. So, the comparison with Wang et al. 2014 is probably a comparison with modelled trunoverrates?

We removed the sentenced as it was confusing.

P13L10 This is confusing: The different parametrisation of Eq. 1 results from data of Bakker et al. (2004) but the sites used here are from Van Oost et al. 2007. So, how to know that these sites fit better to on of the values of B in Eq. 1.

We tried to estimate the relationship based on information from the original studies and from Van Oost et al. (2007). Please refer to the comment above for more details (P9L6). We modified the manuscript as the attribution of a functional form to a specific is subject to a high uncertainty.

5   P19L11 I suggest to look into the leaching data from the group at ZALF Müncheberg. The leaching aspect under erosion seemed to be quite complex. (More DOC leaching under erosion as less evapotranspiration ?)

We clarified and discussed that we did not include processes affecting the vertical distribution of SOC in the profile (bioturbation, C leaching,and tissue allocation). We discussed this and the potential effect of erosion on C leaching 10   and the relative strength compared to the vertical C fluxes.

[revised manuscript text omitted]

---

## Author Response (AR3)

Reply to the Editors

Dear Authors,

5    First of all, I would like to thank you for choosing SOIL as an outlet for your work! Also, thank you for working with the reviewers and editor on improving this manuscript. I am happy to accept this manuscript now, given that you indeed will take care of the typos that remain in the text.

best regards,
10    Jo

Dear authors,
thank you for the detailed response to my last comments. I think the paper has substantially improved, and is now from my side ready to be published. There are only a small number of typos left to be correct.
15    Best regards
Peter Fiener

We thank the editors for these positive comments and for accepting this paper. We have worked on the manuscript to correct the typos. You will find it in the track changed version below.
20
Best regards,
The authors.

[revised manuscript text omitted]